# Synthesis of Oleanolic Acid-Dithiocarbamate Conjugates and Evaluation of Their Broad-Spectrum Antitumor Activities

**DOI:** 10.3390/molecules28031414

**Published:** 2023-02-02

**Authors:** Liyao Tang, Yan Zhang, Jinrun Xu, Qingfan Yang, Fukuan Du, Xu Wu, Mingxing Li, Jing Shen, Shuai Deng, Yueshui Zhao, Zhangang Xiao, Yu Chen

**Affiliations:** 1Laboratory of Molecular Pharmacology, Department of Pharmacology, School of Pharmacy, Southwest Medical University, Luzhou 646000, China; 2Department of Oncology, The Affiliated Hospital of Southwest Medical University, Southwest Medical University, Luzhou 646000, China; 3Cell Therapy & Cell Drugs of Luzhou Key Laboratory, Southwest Medical University, Luzhou 646000, China

**Keywords:** structural modification, oleanolic acid, natural product derivative, dithiocarbamate, hybrid strategy, antitumor activity

## Abstract

Efficient and mild synthetic routes for bioactive natural product derivatives are of current interest for drug discovery. Herein, on the basis of the pharmacophore hybrid strategy, we report a two-step protocol to obtain a series of structurally novel oleanolic acid (OA)-dithiocarbamate conjugates in mild conditions with high yields. Moreover, biological evaluations indicated that representative compound **3e** exhibited the most potent and broad-spectrum antiproliferative effects against Panc1, A549, Hep3B, Huh-7, HT-29, and Hela cells with low cytotoxicity on normal cells. In terms of the IC_50_ values, these OA-dithiocarbamate conjugates were up to 30-fold more potent than the natural product OA. These compounds may be promising hit compounds for the development of novel anti-cancer drugs.

## 1. Introduction

Natural products and their derivatives have a long history in cancer therapy and are important for drug development. Efficient and mild synthetic routes for bioactive natural product derivatives are of current interest for drug discovery [1,2,3,4]. Recently, pentacyclic triterpenes have been identified as the main biologically active components in many traditional Chinese medicines [5,6]. Among them, oleanolic acid (OA) is the most abundant and cheap; thus, OA and its derivatives have been widely investigated for their diverse biological activities, including their anti-cancer, anti-inflammatory, anti-HIV, anti-bacterial, anti-diabetic, and anti-hepatotoxic effects, among others [7,8,9,10,11]. Derivatization of OA has yielded a wide variety of novel compounds for anti-cancer investigations (Figure 1) [11,12,13,14,15]; however, poor pharmacokinetic properties, low cell selectivities, limited bioavailabilities, and synthetic complexity have hindered further clinical application [7]. Therefore, methods for readily accessible modification of OA to enhance its polarity and anti-proliferative activity are urgently required.

Dithiocarbamates are an important class of sulfur-containing organic compounds with a wide range of applications in both academia and industry [16,17,18,19,20,21,22,23,24,25,26,27]. They serve as fungicides and pesticides in agriculture [17,18,19], vulcanization agents in the rubber industry [20], radical chain transfer agents in polymerization [21], effective ligands in coordination chemistry [22], and, last but not least, as biologically important structural motifs in medicinal chemistry (Figure 2) [23,24,25,26,27].

In recent years, the pharmacophore hybrid strategy has emerged as an essential method for the discovery and modification of lead compounds [28,29,30,31]. Covalently combining two known pharmacophores yields a novel hybrid molecule, which can possess integrated advantages for optimizing certain biological activities and overcoming the deficiencies of a single drug [32,33,34,35]. In view of the high performance of dithiocarbamate derivatives in structural modification, the synthesis of OA-dithiocarbamate conjugates may enhance the polarities and antitumor properties of the reaction products in a readily accessible manner [7,23,24,25,26,27]. The structural modifications of OA have mainly focused on the C-3 hydroxyl and C-28 carboxyl groups (Figure 1) [7]. The C-28 carboxyl group can easily be esterified by alcohols or amidated by amines; however, the preparation of OA-dithiocarbamate conjugates has not yet been documented in the literature [7,8,9,10,11]. In order to simplify the synthetic route and control the polarity of target molecules, ethylidene was chosen as a linker between OA and dithiocarbamates.

## 2. Results and Discussion

To establish the optimal reaction conditions, we prepared key intermediate **2,** as previously described [36,37]. Under the “standard” conditions, the reaction of **2** with CS_2_ and pyrrolidine in a one-pot manner afforded the target product **3a** in an 80% isolated yield. In the “standard” conditions, 2 equiv. of K_3_PO_4_ was shown to be essential to yield the desired product **3a** (Entries 1–4, Table 1). Lowering the loading of K_3_PO_4_ to 1.5 equiv. led to a decreased yield of **3a** (Entry 1, Table 1), while replacement of it by K_2_HPO_4_ or Li_2_CO_3_ resulted in no desired product (Entries 2–3, Table 1). On the other hand, in the presence of 2 equiv. of K_2_CO_3_, product **3a** could be isolated with a 62% yield (Entry 4, Table 1). Changing the reaction temperature or using other solvents, such as DMF, CH_3_CN, and EtOH, did not offer better results (Entries 5–8, Table 1). Lower amounts of CS_2_ or pyrrolidine resulted in a decreased yield of **3a** (Entries 9–10, Table 1).

With the optimal reaction conditions in hand, the substrate scope was subsequently investigated, and the results are compiled in Figure 1. The replacement of the H-atom of the pyrrolidine ring with other substituents, such as methyl, dimethyl, hydroxy, and hydroxymethyl, worked well, affording the corresponding products **3b**–**3e** in 69–85% yields. Among them, hydroxyl containing products were obtained at slightly lower yields. This reaction was also tolerant of fused-ring substrates, such as hexahydroisoindoline and isoindoline, resulting in **3f** and **3g** with 77% and 90% yields, respectively.

To further enhance the structural diversity of products, various types of piperidine-derived substrates were also examined, and all of them were compatible with the established reaction conditions. First, methyl-, hydroxy-, hydroxymethyl-, hydroxyethyl-, and phenyl-substituted piperidines reacted smoothly to give **3h**–**3m** in 70–88% yields. Then, methyl-, hydroxyethyl-, phenyl-, and aryl-substituted piperazines were also viable substrates, affording **3n**–**3s** in 71–89% yields. Moreover, thiomorpholine was also compatible, leading to the formation of **3t** in 72% yield. Gratifyingly, the mild reaction conditions, high yields of products, and good functional group tolerances clearly demonstrated the advantages of our pharmacophore hybrid strategy for the structural modification of OA. The isolated compounds **3a**–**3t** were fully characterized by ^1^H and ^13^C NMR spectroscopy as well as high-resolution mass spectrometry (see the Appendix A for details).

Having obtained a series of structurally diverse OA-dithiocarbamates, we next performed a systematic biological evaluation to examine whether introducing an extra dithiocarbamate group could improve antitumor activities. These compounds were evaluated by MTT assay against human pancreatic cancer (Panc1), human lung cancer (A549), human hepatoma cell (Hep3B), human hepatoma cell (Huh-7), human colon cancer (HT-29), and human cervical cancer (Hela) cells, with the widely used anticancer drugs fluorouracil, docetaxel, and cisplatin as positive controls (Table 2). Most of the compounds exhibited remarkable antiproliferative activities, and the IC_50_ values of ten selected compounds were less than 50 µM on certain tumor cell lines. Among them, compounds **3e**, **3i**, **3j,** and **3l** were shown to be excellent, with broad-spectrum antitumor activities as well as being up to 30-fold more potent than the natural product OA and the positive controls; this might be ascribed to the introduction of hydroxyl groups. Particularly, compound **3p** was also found to be a promising hit compound that was 20-fold more potent than the natural product OA against HT-29 cells. Moreover, the cytotoxicities of compounds **3a**-**3t** were also evaluated in human normal hepatocytes (LO2) to determine whether these compounds preferred killing tumor cells over normal cells. Excitingly, the IC_50_ value of compound **3e** in LO2 cells was 62.8 µM, which was several times higher than that in the tumor cells.

## 3. Materials and Methods

### 3.1. General Information

All organic solvents were dried and distilled by standard methods prior to use. ^1^H and ^13^C NMR spectra were recorded on a Bruker AV II-400 spectrometer (BURKERT, Ingelfingen, Germany) at 400 and 100 MHz, respectively. All chemical shifts were reported in δ units with references to the residual solvent resonances of the deuterated solvents for proton and carbon chemical shifts. High Resolution Mass Spectra (HRMS) were obtained on a Thermo Q Exactive™ Focus Hybrid Quadrupole-Orbitrap™ Mass Spectrometer (SCIEX, Framingham, Massachusetts, USA). All other chemicals were purchased from either Aldrich (Sigma-Aldrich, Shanghai, China) or Aladdin Chemical Co. (Aladdin Holdings Group Co., Ltd, Shanghai, China) and used as received, unless otherwise specified.

The optical density at 490 nm of each well was measured using a microplate reader (Molecular devices corporation, Sunnyvale, CA. USA) to calculate the percent of cell viability. The inhibition rates were calculated using GraphPad Prism 7.0 software. The seven tested cell lines were obtained from the laboratory of Molecular Pharmacology, Department of Pharmacology, School of Pharmacy, Southwest Medical University.

### 3.2. Experimental Section of Synthesis

[2-bromoethyl] 3-hydroxy-12-en-28-oic acid (**2**) [36,37,38] To a mixture of oleanolic acid (913.4 mg, 2.0 mmol), K_2_CO_3_ (552.8 mg, 4.0 mmol), and DMF (40 mL), 1, 2-dibromoethane (513 µL, 6.0 mmol) was slowly added at room temperature, and the mixture was then stirred at 40 °C for 4 h. The resulting mixture was cooled to room temperature, then quenched with ice water (50.0 mL), and the insoluble material was removed by a Buchner funnel. The organic layer was separated, and the aqueous layer was extracted with ethyl acetate (50 mL × 3). The organic solutions were combined and dried over anhydrous MgSO_4_. After removal of the solvent, the residue was submitted to column chromatography on silica gel (200-300 mesh) using petroleum ether and ethyl acetate (15/1 in *v*/*v*) as eluents to give **2** (957.6 mg, 85% yield) as a white solid. ^1^H NMR (400 MHz, CDCl_3_): δ 5.30 (s, 1H, H-12), 4.35 (m, 2H, -OC*H*_2_C-), 3.49 (t, *J* = 5.5 Hz, 2H, BrC*H*_2_C-), 3.20 (d, *J* = 6.9 Hz, 1H, H-3), 2.87 (d, *J* = 12.2 Hz, 1H, H-18), 1.99 (m, 1H, -O*H*), 1.87 (m, 2H, -C*H*_2_), 1.72 (m, 3H, H-22, -C*H*, -C*H*_2_,), 1.62 (m, 6H, 3 × -C*H*_2_), 1.54 (m, 3H, H-22, -C*H*, -C*H*_2_), 1.33 (m, 6H, 3 × -C*H*_2_), 1.18 (s, 1H, H-9), 1.13 (s, 3H, -C*H*_3_), 1.06 (s, 1H, H-5), 0.98 (s, 3H, -C*H*_3_), 0.93 (s, 3H, -C*H*_3_), 0.90 (s, 6H, 2 × -C*H*_3_), 0.77 (s, 3H, -C*H*_3_), 0.73 (s, 3H, -C*H*_3_). ^13^C NMR (100 MHz, CDCl_3_): δ 177.2 (C-28), 143.4 (C-13), 122.6 (C-12), 78.8 (C-3), 63.6 (-*C*O-), 55.2 (C-5), 47.6 (C-9), 46.8 (C-17), 45.7 (C-19), 41.6 (C-14), 41.2 (C-18), 39.3 (C-8), 38.7 (C-1), 38.4 (C-4), 37.0 (C-10), 33.8 (C-29), 33.1 (C-22), 32.7 (C-21), 32.4 (C-7), 30.7 (C-20), 29.1 (-*C*Br), 28.2 (C-15), 27.7 (C-23), 27.1 (C-27), 25.9 (C-30), 23.6 (C-2), 23.4 (C-11), 22.9 (C-16), 18.3 (C-6), 17.0 (C-26), 15.6 (C-24), 15.3 (C-25). HRMS (ESI): *m*/*z* calculated for C_32_H_51_BrO_3_ [M+H]^+^: 563.3100. Found: 563.3054.

[2-((pyrrolidine-1-carbonothioyl)thio)ethyl] 3-hydroxy-12-en-28-oic acid (**3a**). To a mixture of CS_2_ (1.8 mmol, 108 µL), anhydrous K_3_PO_4_ (0.8 mmol, 169.1 mg), and THF (8.0 mL), pyrrolidine (1.0 mmol, 82 µL) was slowly added at 0 °C, and the reaction mixture was then stirred at 0 °C for 0.5 h. To the resulting mixture another THF solution (4.0 mL) of **2** (0.4 mmol, 225.4 mg) was added dropwise. The reaction mixture was stirred for 12 h at room temperature, then quenched with ice water (15.0 mL), and the insoluble material was removed by a Buchner funnel. After removal of the solvent, the residue was dissolved in ethyl acetate (15.0 mL). Water (15.0 mL) was added to the resulting solution, the organic layer was separated, and the aqueous layer was extracted with ethyl acetate (15.0 mL × 2). The organic solutions were combined and dried over anhydrous Na_2_SO_4_. After removal of the solvent, the residue was submitted to column chromatography on silica gel (200–300 mesh) using petroleum ether and ethyl acetate (2/1 in *v*/*v*) as eluents to give **3a** (201.3 mg, 80% yield) as a white solid. ^1^H NMR (400 MHz, CDCl_3_): δ 5.29 (s, 1H, H-12), 4.27 (m, 2H, -OC*H*_2_C-), 3.92 (t, *J* = 6.9 Hz, 2H, -SC*H*_2_), 3.66 (t, *J* = 6.8 Hz, 2H, -NC*H*_2_), 3.59 (m, 2H, -NC*H*_2_), 3.20 (m, 1H, H-3), 2.87 (dd, *J* = 13.5, 3.4 Hz, 1H, H-18), 2.10 (m, 1H, -O*H*), 1.97 (m, 5H, H-22, -C*H*, 2 × -C*H*_2_), 1.87 (m, 2H, -C*H*_2_), 1.66 (d, *J* = 8.4 Hz, 3H, -NCH_2_C***H***_2_C***H***), 1.59 (m, 5H, -NCH_2_CH_2_C***H***, 2 × -C*H*_2_), 1.53 (m, 3H, H-22, -C*H*, -C*H*_2_), 1.35 (m, 6H, 3 × -C*H*_2_), 1.16 (d, *J* = 4.0 Hz, 1H, H-9), 1.13 (s, 3H, -C*H*_3_), 1.04 (s, 1H, H-5), 0.98 (s, 3H, -C*H*_3_), 0.93 (s, 3H, -C*H*_3_), 0.90 (s, 6H, 2 × -C*H*_3_), 0.77 (s, 3H, -C*H*_3_), 0.73 (s, 3H, -C*H*_3_). ^13^C NMR (100 MHz, CDCl_3_): δ 191.7 (-*C*S_2_), 177.4 (C-28), 143.6 (C-13), 122.5 (C-12), 78.8 (C-3), 62.5 (-*C*O-), 55.2 (-N*C*H_2_), 55.1 (-N*C*H_2_), 50.6 (C-5), 47.6 (C-9), 46.7 (C-17), 45.8 (C-19), 41.6 (C-14), 41.2 (C-18), 39.3 (C-8), 38.7 (C-1), 38.5 (C-4), 37.0 (C-10), 35.0 (-*C*S), 33.8 (C-29), 33.1 (C-22), 32.7 (C-21), 32.4 (C-7), 30.7 (C-20), 28.1 (C-15), 27.7 (C-23), 27.2 (C-27), 26.1 (-*C*H_2_), 25.9 (C-30), 24.3 (-*C*H_2_), 23.7 (C-2), 23.4 (C-11), 22.9 (C-16), 18.3 (C-6), 17.1 (C-26), 15.7 (C-24), 15.3 (C-25). HRMS (ESI): *m*/*z* calculated for C_37_H_59_NO_3_S_2_ [M+H]^+^: 630.4015. Found: 630.3961.

[2-((2-methylpyrrolidine-1-carbonothioyl)thio)ethyl] 3-hydroxy-12-en-28-oic acid (**3b**). To a mixture of CS_2_ (1.8 mmol, 108 µL), anhydrous K_3_PO_4_ (0.8 mmol, 169.1 mg), and THF (8.0 mL), 2-methylpyrrolidine (1.0 mmol, 101 µL) was slowly added at 0 °C, and the reaction mixture was then stirred at 0 °C for 0.5 h. Another THF solution (4.0 mL) of **2** (0.4 mmol, 225.2 mg) was added dropwise to the resulting mixture. The reaction mixture was stirred for 12 h at room temperature, then quenched with ice water (15.0 mL), and the insoluble material was removed by a Buchner funnel. After removal of the solvent, the residue was dissolved in ethyl acetate (15.0 mL). Water (15.0 mL) was added to the resulting solution, the organic layer was separated, and the aqueous layer was extracted with ethyl acetate (15.0 mL × 2). The organic solutions were combined and dried over anhydrous Na_2_SO_4_. After removal of the solvent, the residue was submitted to column chromatography on silica gel (200–300 mesh) using petroleum ether and ethyl acetate (10/1 in *v*/*v*) as eluents to give **3b** (213.6 mg, 83% yield) as a white solid. ^1^H NMR (400 MHz, CDCl_3_): δ 5.30 (s, 1H, H-12), 4.52 (m, 1H, -O*H*), 4.26 (m, 2H, -OC*H*_2_C-), 3.93 (m, 1H, -NC*H*), 3.73 (m, 1H, -NC*H*), 3.44 (m, 2H, -SC*H*_2_), 3.21 (m, 1H, H-3), 2.87 (dd, *J* = 13.7, 3.8 Hz, 1H, H-18), 2.25 (m, 1H, -NC*H*), 2.02 (m, 4H, 2 × -C*H*_2_), 1.81 (m, 3H, H-11, -C*H*, -C*H*_2_), 1.63 (m, 7H,-NCH_2_C***H***_2_, -NCH_2_C***H***_2_, H-22, -C*H*, -C*H*_2_), 1.53 (m, 3H, H-22, -C*H*, -C*H*_2_), 1.42 (m, 2H, -C*H*_2_), 1.35 (m, 4H, 2 × -C*H*_2_), 1.28 (m, 3H, -C*H*_3_), 1.16 (t, *J* = 4.2 Hz, 1H, H-9), 1.13 (s, 3H, -C*H*_3_), 1.04 (s, 1H, H-5), 0.98 (s, 3H, -C*H*_3_), 0.93 (s, 3H, -C*H*_3_), 0.90 (s, 6H, 2 × -C*H*_3_), 0.77 (s, 3H, -C*H*_3_), 0.74 (s, 3H, -C*H*_3_). ^13^C NMR (100 MHz, CDCl_3_): δ 191.7 (-*C*S_2_), 177.5 (C-28), 143.6 (C-13), 122.5 (C-12), 78.9 (C-3), 62.6 (-*C*O-), 61.3 (-N***C***H_2_CH_3_), 58.0(-*C*H_3_), 55.2 (C-5), 50.4 (-N*C*H_2_), 47.6 (C-9), 46.7 (C-17), 45.8 (C-19), 41.7 (C-14), 41.3 (C-18), 39.4 (C-8), 38.8 (C-1), 38.5 (C-4), 37.0 (C-10), 34.8 (-*C*S), 33.9 (C-29), 33.1 (C-22), 32.4 (C-21), 31.3 (C-7), 30.7 (C-20), 28.1 (C-15), 27.7 (C-23), 27.2 (C-27), 25.9 (C-30), 23.7 (C-2), 22.9 (C-11), 21.6 (C-16), 18.6 (-*C*H_2_), 18.3 (C-6), 17.5 (-*C*H_2_), 17.1 (C-26), 15.6 (C-24), 15.4 (C-25). HRMS (ESI): *m*/*z* calculated for C_38_H_61_NO_3_S_2_ [M+H]^+^: 644.4171. Found: 644.4116.

[2-((2,2-dimethylpyrrolidine-1-carbonothioyl)thio)ethyl] 3-hydroxy-12-en-28-oic acid (**3c**). To a mixture of CS_2_ (1.8 mmol, 108 µL), anhydrous K_3_PO_4_ (0.8 mmol, 169.1 mg), and THF (8.0 mL), 2,2-dimethylpyrrolidine (1.0 mmol, 120 µL) was slowly added at 0 °C, and the reaction mixture was then stirred at 0 °C for 0.5 h. Another THF solution (4.0 mL) of **2** (0.4 mmol, 225.9 mg) was added dropwise to the resulting mixture. The reaction mixture was stirred for 12 h at room temperature, then quenched with ice water (15.0 mL), and the insoluble material was removed by a Buchner funnel. After removal of the solvent, the residue was dissolved in ethyl acetate (15.0 mL). To the resulting solution was added water (15.0 mL), the organic layer was separated, and the aqueous layer was extracted with ethyl acetate (15.0 mL × 2). The organic solutions were combined and dried over anhydrous Na_2_SO_4_. After removal of the solvent, the residue was submitted to column chromatography on silica gel (200–300 mesh) using petroleum ether and ethyl acetate (10/1 in *v*/*v*) as eluents to give **3c** (223.5 mg, 85% yield) as a white solid. ^1^H NMR (400 MHz, CDCl_3_): δ 5.30 (s, 1H, H-12), 4.26 (m, 2H, -OC*H*_2_C-), 3.83 (t, *J* = 6.8 Hz, 1H, -NC*H*), 3.56 (m, 2H, -SC*H*_2_), 3.21 (m, 1H, -NC*H*), 2.87 (dd, *J* = 13.6, 3.7 Hz, 1H, H-18), 2.01 (m, 1H, -O*H*), 1.96 (m, 2H, -C*H*_2_), 1.88 (m, 3H, H-22, -C*H*, -C*H*_2_), 1.73 (s, 6H, 3 × -C*H*_2_), 1.63 (m, 8H, 4 × -C*H*_2_), 1.55 (m, 3H, -C*H*_3_), 1.46 (m, 3H, -C*H*_3_), 1.31 (m, 6H, 3 × -C*H*_2_), 1.16 (t, *J* = 5.6 Hz, 1H, H-9), 1.13 (s, 3H, -C*H*_3_), 1.04 (s, 1H, H-5), 0.98 (s, 3H, -C*H*_3_), 0.93 (s, 3H, -C*H*_3_), 0.90 (s, 6H, 2 × -C*H*_3_), 0.78 (s, 3H, -C*H*_3_), 0.73 (s, 3H, -C*H*_3_). ^13^C NMR (100 MHz, CDCl_3_): δ 191.4 (-*C*S_2_), 177.5 (C-28), 143.7 (C-13), 122.5 (C-12), 79.0 (C-3), 69.2 (-N***C***CH_3_CH_3_), 62.7 (-*C*O-), 55.2 (-N*C*H_2_), 53.8 (C-5), 47.6 (C-9), 46.7 (C-17), 45.8 (C-19), 43.3 (-*C*H_2_), 41.7 (C-14), 41.3 (C-18), 39.4 (C-8), 38.8 (C-1), 38.5 (C-4), 37.0 (C-10), 34.7 (-*C*S), 33.9 (C-29), 33.2 (C-22), 32.8 (C-21), 32.4 (C-7), 30.7 (C-20), 28.2 (C-15), 27.7 (C-23), 27.2 (C-27), 26.1 (-*C*H_2_), 25.9 (C-30), 24.8 (-*C*H_2_), 23.7 (C-2), 23.5 (C-11), 22.9 (C-16), 22.1 (-*C*H_2_), 18.4 (C-6), 17.1 (C-26), 15.7 (C-24), 15.4 (C-25). HRMS (ESI): *m*/*z* calculated for C_39_H_63_NO_3_S_2_ [M+H]^+^: 658.4328. Found: 658.4303.

[2-((3-hydroxypyrrolidine-1-carbonothioyl)thio)ethyl] 3-hydroxy-12-en-28-oic acid (**3d**). To a mixture of CS_2_ (1.8 mmol, 108 µL), anhydrous K_3_PO_4_ (0.8 mmol, 169.1 mg), and THF (8.0 mL), 3-hydroxypyrrolidine (1.0 mmol, 81 µL) was slowly added at 0 °C, and the reaction mixture was then stirred at 0 °C for 0.5 h. Another THF solution (4.0 mL) of **2** (0.4 mmol, 224.7 mg) was added dropwise to the resulting mixture. The reaction mixture was stirred for 12 h at room temperature, then quenched with ice water (15.0 mL), and the insoluble material was removed by a Buchner funnel. After removal of the solvent, the residue was dissolved in ethyl acetate (15.0 mL). Water (15.0 mL) was added to the resulting solution, the organic layer was separated, and the aqueous layer was extracted with ethyl acetate (15.0 mL × 2). The organic solutions were combined and dried over anhydrous Na_2_SO_4_. After removal of the solvent, the residue was submitted to column chromatography on silica gel (200–300 mesh) using petroleum ether and ethyl acetate (2/1 in *v*/*v*) as eluents to give **3d** (180.7 mg, 70% yield) as a yellowish gel. ^1^H NMR (400 MHz, CDCl_3_): δ 5.31 (s, 1H, H-12), 4.57 (m, 1H, -O*H*), 4.21 (m, 2H, -OC*H*_2_C-), 4.03 (m, 2H, -SC*H*_2_), 3.82 (m, 2H, -NC*H*_2_), 3.55 (m, 2H, -NC*H*_2_), 3.20 (m, 1H, H-3), 2.86 (d, *J* = 12.2 Hz, 1H, H-18), 2.15 (m, 1H, -O*H*), 2.05 (m, 2H, -C*H*_2_), 1.89 (m, 4H, 2 × -C*H*_2_), 1.67 (m, 6H, 3 × -C*H*_2_), 1.53 (m, 4H, 2 × -C*H*_2_), 1.33 (m, 7H, -C***H***OH, 3 × -C*H*_2_), 1.16 (t, *J* = 3.8 Hz, 1H, H-9), 1.13 (s, 3H, -C*H*_3_), 1.05 (s, 1H, H-5), 0.97 (s, 3H, -C*H*_3_), 0.93 (s, 3H, -C*H*_3_), 0.90 (d, *J* = 2.3 Hz, 6H, 2 × -C*H*_3_), 0.77 (s, 3H, -C*H*_3_), 0.73 (s, 3H, -C*H*_3_). ^13^C NMR (100 MHz, CDCl_3_): δ 192.5 (-*C*S_2_), 177.7 (C-28), 143.5 (C-13), 122.5 (C-12), 79.0 (C-3), 70.7 (-*C*OH), 68.8 (-N***C***H_2_CH), 63.2 (-N*C*H_2_), 62.5 (-*C*O-), 55.2 (C-5), 52.9 (-*C*H_2_), 48.6 (C-7), 47.6 (C-9), 46.8 (C-17), 45.8 (C-19), 41.7 (C-14), 41.3 (C-18), 39.3 (C-8), 38.7 (C-1), 38.5 (C-4), 37.0 (C-10), 35.0 (-*C*S), 33.8 (C-29), 32.9 (C-22), 32.4 (C-21), 30.7 (C-20), 28.1 (C-15), 27.7 (C-23), 27.1 (C-27), 25.9 (C-30), 23.7 (C-2), 23.4 (C-11), 22.9 (C-16), 18.3 (C-6), 17.1 (C-26), 15.7 (C-24), 15.4 (C-25). HRMS (ESI): *m*/*z* calculated for C_37_H_59_NO_4_S_2_ [M+H]^+^: 646.3964. Found: 646.3917.

[2-((3-(hydroxymethyl)pyrrolidine-1-carbonothioyl)thio)ethyl] 3-hydroxy-12-en-28-oic acid (**3e**). To a mixture of CS_2_ (1.8 mmol, 108 µL), anhydrous K_3_PO_4_ (0.8 mmol, 169.1 mg), and THF (8.0 mL), 3-(hydroxymethyl)pyrrolidine (1.0 mmol, 103 µL) was slowly added at 0 °C, and the reaction mixture was then stirred at 0 °C for 0.5 h. Another THF solution (4.0 mL) of **2** (0.4 mmol, 225.8 mg) was added dropwise to the resulting mixture. The reaction mixture was stirred for 12 h at room temperature, then quenched with ice water (15.0 mL), and the insoluble material was removed by a Buchner funnel. After removal of the solvent, the residue was dissolved in ethyl acetate (15.0 mL). Water (15.0 mL) was added to the resulting solution, the organic layer was separated, and the aqueous layer was extracted with ethyl acetate (15.0 mL × 2). The organic solutions were combined and dried over anhydrous Na_2_SO_4_. After removal of the solvent, the residue was submitted to column chromatography on silica gel (200–300 mesh) using petroleum ether and ethyl acetate (1/1 in *v*/*v*) as eluents to give **3e** (181.9 mg, 69% yield) as a yellowish gel. ^1^H NMR (400 MHz, CDCl_3_): δ 5.30 (t, *J* = 3.2 Hz, 1H, H-12), 4.25 (m, 2H, -OC*H*_2_C-), 3.85 (m, 1H, -O*H*), 3.68 (m, 2H, -SC*H*_2_), 3.54 (m, 3H, -C***H***_2_OH, -NC*H*), 3.21 (m, 1H, H-3), 2.86 (dd, *J* = 13.7, 3.9 Hz, 1H, H-18), 2.55 (m, 1H, -NC*H*), 2.22 (m, 1H, -NC*H*), 2.08 (m, 1H, -NC*H*), 1.95 (m, 2H, -C*H*_2_), 1.84 (m, 3H, -NCH_2_C***H***, -C*H*_2_), 1.70 (m, 1H, -O*H*), 1.63 (m, 6H, 2 × -C*H*_3_), 1.53 (m, 3H, -NCH_2_C***H***, -C*H*_2_), 1.42 (m, 3H, H-22, -C*H*, -C*H*_2_), 1.28 (m, 5H, H-22, -C*H*, 2 × -C*H*_2_), 1.21 (s, 1H, -C***H***CH_2_OH), 1.17 (t, *J* = 3.2 Hz, 1H, H-9), 1.13 (s, 3H, -C*H*_3_), 1.05 (s, 1H, H-5), 0.98 (s, 3H, -C*H*_3_), 0.93 (s, 3H, -C*H*_3_), 0.90 (s, 6H, 2 × -C*H*_3_), 0.77 (s, 3H, -C*H*_3_), 0.73 (s, 3H, -C*H*_3_). ^13^C NMR (100 MHz, CDCl_3_): δ 192.0 (-*C*S_2_), 177.6 (C-28), 143.6 (C-13), 122.5 (C-12), 79.0 (C-3), 63.6 (-*C*OH), 62.4 (-*C*O-), 57.5 (-N***C***H_2_CH), 55.2 (-*C*`OH), 54.5 (-N***C*`**H_2_CH), 53.1 (-N***C***H_2_CH_2_), 50.1 (C-5), 47.6 (C-9), 46.8 (C-17), 45.8 (C-19), 41.7 (C-14), 41.3 (C-18), 39.5 (-***C***CH_2_OH), 39.3 (C-8), 38.7 (C-1), 38.5 (C-4), 37.0 (C-10), 35.0 (-*C*S), 33.8 (C-29), 33.1 (C-22), 32.7 (C-21), 32.4 (C-7), 31.5 (-N***C*`**H_2_CH_2_), 30.7 (C-20), 30.2 (-NCH_2_***C***H_2_), 28.4 (-***C*`**CH_2_OH), 28.1 (C-15), 27.7 (C-23), 27.1 (C-27), 26.6 (-***C*`**CH_2_OH), 25.9 (C-30), 23.7 (C-2), 23.4 (C-11), 22.9 (C-16), 18.3 (C-6), 17.1 (C-26), 15.7 (C-24), 15.4 (C-25). HRMS (ESI): *m*/*z* calculated for C_38_H_61_NO_4_S_2_ [M+H]^+^: 660.4120. Found: 660.4069.

[2-((octahydro-1H-isoindole-2-carbonothioyl)thio)ethyl] 3-hydroxy-12-en-28-oic acid (**3f**). To a mixture of CS_2_ (1.8 mmol, 108 µL), anhydrous K_3_PO_4_ (0.8 mmol, 169.1 mg), and THF (8.0 mL), octahydro-1H-isoindole (1.0 mmol, 115 µL) was slowly added at 0 °C, and the reaction mixture was then stirred at 0 °C for 0.5 h. Another THF solution (4.0 mL) of **2** (0.4 mmol, 225.5 mg) was added dropwise to the resulting mixture. The reaction mixture was stirred for 12 h at room temperature, then quenched with ice water (15.0 mL), and the insoluble material was removed by a Buchner funnel. After removal of the solvent, the residue was dissolved in ethyl acetate (15.0 mL). Water (15.0 mL) was added to the resulting solution, the organic layer was separated, and the aqueous layer was extracted with ethyl acetate (15.0 mL × 2). The organic solutions were combined and dried over anhydrous Na_2_SO_4_. After removal of the solvent, the residue was submitted to column chromatography on silica gel (200–300 mesh) using petroleum ether and ethyl acetate (10/1 in *v*/*v*) as eluents to give **3f** (210.4 mg, 77% yield) as a yellowish gel. ^1^H NMR (400 MHz, CDCl_3_): δ 5.20 (s, 1H, H-12), 4.18 (m, 2H, -OC*H*_2_C-), 3.76 (m, 2H, -SC*H*_2_), 3.52 (m, 4H, 2 × -NC*H*_2_), 3.10 (m, 1H, H-3), 2.77 (d, *J* = 11.0 Hz, 1H, H-18), 2.26 (m, 2H, -C*H*_2_), 2.09 (m, 1H, -O*H*), 1.84 (m, 3H, H-22, -C*H*, -C*H*_2_), 1.52 (m, 8H, 4 × -C*H*_2_), 1.44 (m, 5H, H-22, -C*H*, 2 × -C*H*_2_), 1.30 (m, 8H, 4 × -C*H*_2_), 1.18 (m, 4H, 2 × -C*H*_2_), 1.07 (s, 1H, H-9), 1.04 (s, 3H, -C*H*_3_), 0.95 (s, 1H, H-5), 0.88 (s, 3H, -C*H*_3_), 0.83 (s, 3H, -C*H*_3_), 0.80 (s, 6H, 2 × -C*H*_3_), 0.67 (s, 3H, -C*H*_3_), 0.64 (s, 3H, -C*H*_3_). ^13^C NMR (100 MHz, CDCl_3_): δ 192.3 (-*C*S_2_), 177.3 (C-28), 143.5 (C-13), 122.5 (C-12), 78.6 (C-3), 62.5 (-*C*O-), 58.9 (2 × -*C*H_2_), 55.2 (C-5), 54.5 (2 × -*C*H_2_), 47.5 (C-9), 46.6 (C-17), 45.7 (C-19), 41.6 (C-14), 41.2 (C-18), 39.3 (C-8), 38.7 (C-1), 37.6 (C-4), 36.9 (C-10), 35.8 (-*C*S), 34.9 (2 × -*C*H_2_), 33.8 (C-29), 33.1 (C-22), 32.7 (C-21), 32.3 (C-7), 30.6 (C-20), 28.1 (C-15), 27.7 (C-23), 27.1 (C-27), 26.9 (C-30), 25.9 (C-2), 25.6 (2 × -*C*H_2_), 23.6 (C-11), 22.6 (C-16), 18.3 (C-6), 17.1 (C-26), 15.7 (C-24), 15.3 (C-25). HRMS (ESI): *m*/*z* calculated for C_41_H_65_NO_3_S_2_ [M+H]^+^: 684.4484. Found: 684.4430.

[2-((isoindoline-2-carbonothioyl)thio)ethyl] 3-hydroxy-12-en-28-oic acid (**3g**). To a mixture of CS_2_ (1.8 mmol, 108 µL), anhydrous K_3_PO_4_ (0.8 mmol, 169.1 mg), and THF (8.0 mL), isoindoline (1.0 mmol, 113 µL) was slowly added at 0 °C, and the reaction mixture was then stirred at 0 °C for 0.5 h. Another THF solution (4.0 mL) of **2** (0.4 mmol, 226.4 mg) was added dropwise to the resulting mixture. The reaction mixture was stirred for 12 h at room temperature, then quenched with ice water (15.0 mL), and the insoluble material was removed by a Buchner funnel. After removal of the solvent, the residue was dissolved in ethyl acetate (15.0 mL). Water (15.0 mL) was added to the resulting solution, the organic layer was separated, and the aqueous layer was extracted with ethyl acetate (15.0 mL × 2). The organic solutions were combined and dried over anhydrous Na_2_SO_4_. After removal of the solvent, the residue was submitted to column chromatography on silica gel (200–300 mesh) using petroleum ether and ethyl acetate (10/1 in *v*/*v*) as eluents to give **3g** (243.9 mg, 90% yield) as a yellowish solid. ^1^H NMR (400 MHz, CDCl_3_): δ 7.31 (m, 4H, Ar-*H*), 5.30 (t, *J* = 3.3 Hz, 1H, H-12), 5.20 (s, 2H, -OC*H*_2_C-), 4.99 (s, 2H, -NC*H*_2_), 4.32 (m, 2H, -NC*H*_2_), 3.66 (m, 2H, -SC*H*_2_), 3.19 (m, 1H, H-3), 2.95 (m, 1H, -O*H*), 2.88 (dd, *J* = 13.7, 4.0 Hz, 1H, H-18), 1.85 (m, 2H, -C*H*_2_), 1.69 (m, 3H, H-22, -C*H*, -C*H*_2_), 1.54 (m, 6H, 3 × -C*H*_2_), 1.42 (m, 3H, H-22, -C*H*, -C*H*_2_), 1.23 (m, 6H, 3 × -C*H*_2_), 1.17 (t, *J* = 4.4 Hz, 1H, H-9), 1.12 (s, 3H, -C*H*_3_), 1.04 (s, 1H, H-5), 0.95 (s, 3H, -C*H*_3_), 0.93 (s, 3H, -C*H*_3_), 0.90 (s, 3H, -C*H*_3_), 0.86 (s, 3H, -C*H*_3_), 0.74 (s, 3H, -C*H*_3_), 0.72 (s, 3H, -C*H*_3_). ^13^C NMR (100 MHz, CDCl_3_): δ 192.7 (-*C*S_2_), 177.5 (C-28), 143.6 (C-13), 135.2 (*Ph*), 134.9 (*Ph*), 128.1 (*Ph*), 127.9 (*Ph*), 122.8 (*Ph*), 122.7 (*Ph*), 122.5 (C-12), 78.9 (C-3), 62.3 (-*C*O-), 60.5 (-N*C*H_2_), 55.7 (-N*C*H_2_), 55.1 (C-5), 47.5 (C-9), 46.7 (C-17), 45.8 (C-19), 41.6 (C-14), 41.3 (C-18), 39.3 (C-8), 38.7 (C-1), 38.4 (C-4), 37.0 (C-10), 35.3 (-*C*S), 33.8 (C-29), 33.1 (C-22), 32.7 (C-21), 32.4 (C-7), 30.7 (C-20), 28.1 (C-15), 27.7 (C-23), 27.1 (C-27), 25.9 (C-30), 23.6 (C-2), 23.4 (C-11), 22.9 (C-16), 18.2 (C-6), 17.1 (C-26), 15.5 (C-24), 15.3 (C-25). HRMS (ESI): *m*/*z* calculated for C41H59NO3S2 [M+H]+: 678.4015. Found: 678.3996.

[2-((4-methylpiperidine-1-carbonothioyl)thio)ethyl] 3-hydroxy-12-en-28-oic acid (**3h**). To a mixture of CS_2_ (1.8 mmol, 108 µL), anhydrous K_3_PO_4_ (0.8 mmol, 169.1 mg), and THF (8.0 mL), 4-methylpiperidine (1.0 mmol, 100 µL) was slowly added at 0 °C, and the reaction mixture was then stirred at 0 °C for 0.5 h. Another THF solution (4.0 mL) of **2** (0.4 mmol, 224.6 mg) was added dropwise to the resulting mixture. The reaction mixture was stirred for 12 h at room temperature, then quenched with ice water (15.0 mL), and the insoluble material was removed by a Buchner funnel. After removal of the solvent, the residue was dissolved in ethyl acetate (15.0 mL). Water (15.0 mL) was added to the resulting solution, the organic layer was separated, and the aqueous layer was extracted with ethyl acetate (15.0 mL × 2). The organic solutions were combined and dried over anhydrous Na_2_SO_4_. After removal of the solvent, the residue was submitted to column chromatography on silica gel (200–300 mesh) using petroleum ether and ethyl acetate (10/1 in *v*/*v*) as eluents to give **3h** (199.8 mg, 76% yield) as a white solid. ^1^H NMR (400 MHz, CDCl_3_): δ 5.23 (s, 1H, H-12), 4.21 (m, 2H, -OC*H*_2_C-), 3.54 (m, 2H, -SC*H*_2_), 3.10 (m, 3H, -NC*H*_2,_ H-3), 2.80 (d, *J* = 12.2 Hz, 1H, H-18), 1.89 (m, 1H, -O*H*), 1.80 (m, 2H, -NC*H*_2_), 1.70 (m, 3H, H-22, -C*H*, -C*H*_2_), 1.58 (m, 7H, -C***H***CH_3_, 3 × -C*H*_2_), 1.46 (m, 4H, 2 × -C*H*_2_), 1.33 (m, 5H, H-22, -C*H*, 2 × -C*H*_2_), 1.20 (m, 6H, 3 × -C*H*_2_), 1.10 (s, 1H, H-9), 1.06 (s, 3H, -C*H*_3_), 0.98 (s, 1H, H-5), 0.92 (s, 6H, 2 × -C*H*_3_), 0.86 (s, 3H, -C*H*_3_), 0.83 (s, 6H, 2 × -C*H*_3_), 0.71 (s, 3H, -C*H*_3_), 0.67 (s, 3H, -C*H*_3_). ^13^C NMR (100 MHz, CDCl_3_): δ 194.8 (-*C*S_2_), 177.5 (C-28), 143.7 (C-13), 122.6 (C-12), 79.1 (C-3), 62.5 (-*C*O-), 55.3 (C-5), 53.6 (-N*C*H_2,_ -N*C*H_2_), 47.7 (C-9), 46.8 (C-17), 45.9 (C-19), 41.8 (C-14), 41.4 (C-18), 39.4 (C-8), 38.8 (C-1), 38.5 (C-4), 37.1 (C-10), 35.8 (-*C*S), 34.0 (C-29), 33.2 (-NCH_2_***C***H_2,_ -NCH_2_***C***H_2_), 33.1 (C-22), 32.8 (C-21), 32.5 (C-7), 31.0 (-***C***CH_3_), 30.8 (C-20), 28.2 (C-15), 27.8 (C-23), 27.3 (C-27), 26.0 (C-30), 23.7 (C-2), 23.5 (C-11), 23.0 (C-16), 21.4 (-*C*H_3_), 18.4 (C-6), 17.2 (C-26), 15.7 (C-24), 15.4 (C-25). HRMS (ESI): *m*/*z* calculated for C_39_H_63_NO_3_S_2_ [M+H]^+^: 658.4328. Found: 658.4275.

[2-((4-hydroxypiperidine-1-carbonothioyl)thio)ethyl] 3-hydroxy-12-en-28-oic acid (**3i**). To a mixture of CS_2_ (1.8 mmol, 108 µL), anhydrous K_3_PO_4_ (0.8 mmol, 169.1 mg), and THF (8.0 mL), 4-hydroxypiperidine (1.0 mmol, 103.5 mg) was slowly added at 0 °C, and the reaction mixture was then stirred at 0 °C for 0.5 h. Another THF solution (4.0 mL) of **2** (0.4 mmol, 225.4 mg) was added dropwise to the resulting mixture. The reaction mixture was stirred for 12 h at room temperature, then quenched with ice water (15.0 mL), and the insoluble material was removed by a Buchner funnel. After removal of the solvent, the residue was dissolved in ethyl acetate (15.0 mL). Water (15.0 mL) was added to the resulting solution, the organic layer was separated, and the aqueous layer was extracted with ethyl acetate (15.0 mL × 2). The organic solutions were combined and dried over anhydrous Na_2_SO_4_. After removal of the solvent, the residue was submitted to column chromatography on silica gel (200–300 mesh) using petroleum ether and ethyl acetate (1/1 in *v*/*v*) as eluents to give **3i** (184.6 mg, 70% yield) as a yellow gel. ^1^H NMR (400 MHz, CDCl_3_): δ 5.50 (s, 1H, -O*H*), 5.30 (s, 1H, H-12), 4.60 (s, 1H, -C***H***OH), 4.28 (m, 2H, -SC*H*_2_), 3.61 (m, 2H, -NC*H*_2_), 3.21 (m, 1H, H-3), 3.08 (s, 2H, -NC*H*_2_), 2.87 (d, *J* = 11.0 Hz, 1H, H-18), 2.00 (m, 1H, -O*H*), 1.87 (s, 2H, -C*H*_2_), 1.76 (m, 3H, H-22, -C*H*, -C*H*_2_), 1.62 (m, 6H, 3 × -C*H*_2_), 1.53 (m, 5H, H-22, -C*H*, 2 × -C*H*_2_), 1.42 (m, 4H, 2 × -C*H*_2_), 1.27 (m, 6H, 3 × -C*H*_2_), 1.16 (t, *J* = 3.7 Hz, 1H, H-9), 1.13 (s, 3H, -C*H*_3_), 1.04 (s, 1H, H-5), 0.98 (s, 3H, -C*H*_3_), 0.93 (s, 3H, -C*H*_3_), 0.90 (s, 6H, 2 × -C*H*_3_), 0.77 (s, 3H, -C*H*_3_), 0.74 (s, 3H, -C*H*_3_). ^13^C NMR (100 MHz, CDCl_3_): δ 194.6 (-*C*S_2_), 177.4 (C-28), 143.6 (C-13), 122.5 (C-12), 78.9 (C-3), 62.4 (-*C*O-), 55.2 (C-5), 47.6 (C-9), 46.7 (C-17), 45.8 (C-19), 41.7 (C-14), 41.3 (C-18), 39.4 (C-8), 38.8 (C-1), 38.5 (C-4), 37.0 (C-10), 35.7 (-*C*S), 33.9 (C-29), 33.1 (C-22), 32.8 (C-21), 32.4 (C-7), 31.0 (-*C*OH), 30.7 (C-20), 28.1 (C-15), 27.7 (C-23), 27.2 (C-27), 26.9 (-N*C*H_2,_ -N*C*H_2_), 25.9 (C-30), 23.7 (C-2), 23.4 (C-11), 22.9 (C-16), 21.3 (-NCH_2_***C***H_2,_ -NCH_2_***C***H_2_), 18.4 (C-6), 17.1 (C-26), 15.6 (C-24), 15.4 (C-25). HRMS (ESI): *m*/*z* calculated for C_38_H_61_NO_4_S_2_ [M+H]^+^: 660.4120. Found: 660.4080.

[2-((4-(hydroxymethyl)piperidine-1-carbonothioyl)thio)-ethyl] 3-hydroxy-12-en-28-oic acid (**3j**). To a mixture of CS_2_ (1.8 mmol, 108 µL), anhydrous K_3_PO_4_ (0.8 mmol, 169.1 mg), and THF (8.0 mL), 4-(hydroxymethyl)piperidine (1.0 mmol, 115.3 mg) was slowly added at 0 °C, and the reaction mixture was then stirred at 0 °C for 0.5 h. Another THF solution (4.0 mL) of **2** (0.4 mmol, 225.9 mg) was added dropwise to the resulting mixture. The reaction mixture was stirred for 12 h at room temperature, then quenched with ice water (15.0 mL), and the insoluble material was removed by a Buchner funnel. After removal of the solvent, the residue was dissolved in ethyl acetate (15.0 mL). Water (15.0 mL) was added to the resulting solution, the organic layer was separated, and the aqueous layer was extracted with ethyl acetate (15.0 mL × 2). The organic solutions were combined and dried over anhydrous Na_2_SO_4_. After removal of the solvent, the residue was submitted to column chromatography on silica gel (200–300 mesh) using petroleum ether and ethyl acetate (2/1 in *v*/*v*) as eluents to give **3j** (199.3 mg, 74% yield) as a yellow gel. ^1^H NMR (400 MHz, CDCl_3_): δ 5.57 (s, 1H, -C***H***CH_2_OH), 5.31 (s, 1H, H-12), 4.64 (s, 1H, -C***H***OH), 4.26 (m, 2H, -OC*H*_2_C-), 3.60 (m, 2H, -SC*H*_2_), 3.51 (m, 2H, -NC*H*_2_), 3.21 (m, 2H, -NC*H*_2_), 3.13 (m, 1H, H-3), 2.86 (dd, *J* = 13.5, 3.7 Hz, 1H, H-18), 1.96 (m, 1H, -O*H*), 1.91 (m, 2H, -C*H*_2_), 1.87 (m, 3H, -O*H*, -C*H*_2_), 1.63 (m, 6H, 3 × -C*H*_2_), 1.52 (m, 5H, -C***H***OH, 2 × -CH_2_), 1.38 (m, 6H, 3 × -C*H*_2_), 1.27 (m, 4H, 2 × -C*H*_2_), 1.16 (t, *J* = 4.4 Hz, 1H, H-9), 1.13 (s, 3H, -C*H*_3_), 1.05 (s, 1H, H-5), 0.98 (s, 3H, -C*H*_3_), 0.93 (s, 3H, -C*H*_3_), 0.90 (s, 6H, 2 × -C*H*_3_), 0.77 (s, 3H, -C*H*_3_), 0.73 (s, 3H, -C*H*_3_). ^13^C NMR (100 MHz, CDCl_3_): δ 194.8 (-*C*S_2_), 177.5 (C-28), 143.6 (C-13), 122.5 (C-12), 78.9 (C-3), 66.5 (-*C*H_2_OH), 62.4 (-*C*O-), 55.2 (C-5), 47.6 (C-9), 46.7 (C-17), 45.8 (C-19), 41.6 (C-14), 41.3 (C-18), 39.3 (C-8), 38.7 (C-1), 38.5 (C-4), 38.4 (-N*C*H_2,_ -N*C*H_2_), 37.0 (C-10), 36.6 (-N***C***H_2_CH_2,_ -N***C***H_2_CH_2_), 35.6 (-*C*S), 33.8 (C-29), 33.1 (C-22), 32.7 (C-21), 32.4 (C-7), 31.5 (-***C***CH_2_OH), 30.7 (C-20), 28.1 (C-15), 27.7 (C-23), 27.1 (C-27), 25.9 (C-30), 23.6 (C-2), 23.4 (C-11), 22.9 (C-16), 18.3 (C-6), 17.1 (C-26), 15.6 (C-24), 15.3 (C-25). HRMS (ESI): *m*/*z* calculated for C_39_H_63_NO_4_S_2_ [M+H]^+^: 674.4277. Found: 674.4230.

[2-((2-(2-hydroxyethyl)piperidine-1-carbonothioyl)thio)ethyl] 3-hydroxy-12-en-28-oic acid (**3k**). To a mixture of CS_2_ (1.8 mmol, 108 µL), anhydrous K_3_PO_4_ (0.8 mmol, 169.1 mg), and THF (8.0 mL), 2-(2-hydroxyethyl)piperidine (1.0 mmol, 128.1 mg) was slowly added at 0 °C, and the reaction mixture was then stirred at 0 °C for 0.5 h. Another THF solution (4.0 mL) of **2** (0.4 mmol, 225.6 mg) was added dropwise to the resulting mixture. The reaction mixture was stirred for 12 h at room temperature, then quenched with ice water (15.0 mL), and the insoluble material was removed by a Buchner funnel. After removal of the solvent, the residue was dissolved in ethyl acetate (15.0 mL). Water (15.0 mL) was added to the resulting solution, the organic layer was separated, and the aqueous layer was extracted with ethyl acetate (15.0 mL × 2). The organic solutions were combined and dried over anhydrous Na_2_SO_4_. After removal of the solvent, the residue was submitted to column chromatography on silica gel (200–300 mesh) using petroleum ether and ethyl acetate (2/1 in *v*/*v*) as eluents to give **3k** (200.7 mg, 73% yield) as a yellowish gel. ^1^H NMR (400 MHz, CDCl_3_): δ 5.93 (m, 1H, -O*H*), 5.30 (t, *J* = 3.3 Hz, 1H, H-12), 4.55 (m, 1H, -C***H***OH), 4.28 (m, 2H, -OC*H*_2_C-), 3.62 (m, 3H, -C***H***OH, -SC*H*_2_), 3.39 (m, 1H, H-3), 3.16 (m, 3H, -NC*H*_2_, -NC*H*), 2.87 (dd, *J* = 13.7, 3.9 Hz, 1H, H-18), 2.14 (m, 1H, -O*H*), 1.95 (m, 2H, -C*H*_2_), 1.87 (m, 3H, -CHC***H***_2_CH_2_OH, H-22, -C*H*), 1.79 (m, 1H, H-22, -C*H*), 1.70 (m, 6H, 3 × -C*H*_2_), 1.59 (m, 6H, 3 × -C*H*_2_), 1.53 (m, 4H, 2 × -C*H*_2_), 1.38 (m, 6H, 3 × -C*H*_2_), 1.17 (t, *J* = 4.3 Hz, 1H, H-9), 1.13 (s, 3H, -C*H*_3_), 1.05 (s, 1H, H-5), 0.98 (s, 3H, -C*H*_3_), 0.93 (s, 3H, -C*H*_3_), 0.90 (s, 6H, 2 × -C*H*_3_), 0.77 (s, 3H, -C*H*_3_), 0.74 (s, 3H, -C*H*_3_). ^13^C NMR (100 MHz, CDCl_3_): δ 196.4 (-*C*S_2_), 177.4 (C-28), 143.5 (C-13), 122.5 (C-12), 78.8 (C-3), 58.1 (-*C*O-), 56.0 (-N*C*H), 55.2 (C-5), 47.6 (C-9), 46.7 (C-17), 46.0 (C-19), 45.8 (-*C*H_2_OH), 41.6 (C-14), 41.3 (C-18), 39.3 (C-8), 38.7 (C-1), 38.5 (C-4), 37.0 (C-10), 35.6 (-*C*S), 33.8 (C-29), 33.1 (C-22), 32.9 (C-21), 32.7 (C-7), 32.4 (-N*C*H_2_), 30.7 (C-20), 29.3 (-***C***H_2_CH_2_OH), 28.1 (C-15), 27.7 (C-23), 27.1 (C-27), 25.9 (C-30), 25.8 (-NCH***C***H_2_), 23.6 (C-2), 23.4 (C-11), 22.9 (C-16), 19.2 (-NCH_2_CH_2_***C***H_2_), 18.3 (C-6), 17.1 (C-26), 17.1 (-NCH_2_***C***H_2_), 15.6 (C-24), 15.3 (C-25). HRMS (ESI): *m*/*z* calculated for C_40_H_65_NO_4_S_2_ [M+H]^+^: 688.4433. Found: 688.4418.

[2-((4-(2-hydroxyethyl)piperidine-1-carbonothioyl)thio)ethyl] 3-hydroxy-12-en-28-oic acid (**3l**). To a mixture of CS_2_ (1.8 mmol, 108 µL), anhydrous K_3_PO_4_ (0.8 mmol, 169.1 mg), and THF (8.0 mL), 4-(2-hydroxyethyl)piperidine (1.0 mmol, 130.8 mg) was slowly added at 0 °C, and the reaction mixture was then stirred at 0 °C for 0.5 h. Another THF solution (4.0 mL) of **2** (0.4 mmol, 226.4 mg) was added dropwise to the resulting mixture. The reaction mixture was stirred for 12 h at room temperature, then quenched with ice water (15.0 mL), and the insoluble material was removed by a Buchner funnel. After removal of the solvent, the residue was dissolved in ethyl acetate (15.0 mL). Water (15.0 mL) was added to the resulting solution, the organic layer was separated, and the aqueous layer was extracted with ethyl acetate (15.0 mL × 2). The organic solutions were combined and dried over anhydrous Na_2_SO_4_. After removal of the solvent, the residue was submitted to column chromatography on silica gel (200–300 mesh) using petroleum ether and ethyl acetate (2/1 in *v*/*v*) as eluents to give **3l** (195.2 mg, 71% yield) as a yellowish gel. ^1^H NMR (400 MHz, CDCl_3_): δ 5.23 (t, *J* = 3.3 Hz, 1H, H-12), 4.53 (m, 1H, -O*H*), 4.21 (m, 2H, -OC*H*_2_C-), 3.95 (m, 2H, -SC*H*_2_), 3.75 (m, 2H, -NC*H*_2_), 3.53 (m, 2H, -NC*H*_2_), 3.14 (m, 1H, H-3), 2.79 (dd, *J* = 13.6, 3.7 Hz, 1H, H-18), 2.07 (m, 2H, -C*H*_2_), 1.90 (m, 1H, -O*H*), 1.81 (m, 3H, -C*H*_2_, -NCH_2_CH_2_C***H***), 1.61 (m, 2H, -C*H*_2_), 1.56 (m, 6H, 3 × -C*H*_2_), 1.45 (m, 4H, 2 × -C*H*_2_), 1.25 (m, 6H, 3 × -C*H*_2_), 1.10 (t, *J* = 4.6 Hz, 1H, H-9), 1.06 (s, 3H, -C*H*_3_), 0.98 (s, 1H, H-5), 0.91 (s, 3H, -C*H*_3_), 0.86 (s, 3H, -C*H*_3_), 0.83 (s, 6H, 2 × -C*H*_3_), 0.70 (s, 3H, -C*H*_3_), 0.67 (s, 3H, -C*H*_3_). ^13^C NMR (100 MHz, CDCl_3_): δ 194.5 (-*C*S_2_), 177.5 (C-28), 143.5 (C-13), 122.5 (C-12), 77.0 (C-3), 62.4 (-*C*O-), 60.4 (-*C*H_2_OH), 59.6 (-N*C*H_2_), 55.2 (C-5), 52.3 (-N*C*H_2_), 50.4 (-***C***H_2_CH_2_OH), 47.5 (C-9), 46.7 (C-17), 45.7 (C-19), 41.6 (C-14), 41.2 (C-18), 39.3 (C-8), 38.7 (C-1), 38.5 (C-4), 36.9 (C-10), 35.6 (-CS), 33.8 (C-29), 33.1 (C-22), 32.7 (C-21), 32.4 (C-7), 31.6 (-NCH_2_***C***H_2_), 30.7 (C-20), 28.1 (C-15), 27.6 (C-23), 27.1 (C-27), 25.9 (C-30), 23.6 (C-2), 23.4 (C-11), 22.9 (C-16), 21.1 (-***C***HCH_2_CH_2_OH), 18.3 (C-6), 17.1 (C-26), 15.7 (C-24), 15.3 (C-25). HRMS (ESI): *m*/*z* calculated for C_40_H_65_NO_4_S_2_ [M+H]^+^: 688.4433. Found: 688.4366.

[2-((4-phenylpiperidine-1-carbonothioyl)thio)ethyl] 3-hydroxy-12-en-28-oic acid (**3m**). To a mixture of CS_2_ (1.8 mmol, 108 µL), anhydrous K_3_PO_4_ (0.8 mmol, 169.1 mg), and THF (8.0 mL), 4-phenylpiperidine (1.0 mmol, 164.7 mg) was slowly added at 0 °C, and the reaction mixture was then stirred at 0 °C for 0.5 h. Another THF solution (4.0 mL) of **2** (0.4 mmol, 225.2 mg) was added dropwise to the resulting mixture. The reaction mixture was stirred for 12 h at room temperature, then quenched with ice water (15.0 mL), and the insoluble material was removed by a Buchner funnel. After removal of the solvent, the residue was dissolved in ethyl acetate (15.0 mL). Water (15.0 mL) was added to the resulting solution, the organic layer was separated, and the aqueous layer was extracted with ethyl acetate (15.0 mL × 2). The organic solutions were combined and dried over anhydrous Na_2_SO_4_. After removal of the solvent, the residue was submitted to column chromatography on silica gel (200–300 mesh) using petroleum ether and ethyl acetate (10/1 in *v*/*v*) as eluents to give **3m** (253.2 mg, 88% yield) as a yellowish gel. ^1^H NMR (400 MHz, CDCl_3_): δ 7.31 (m, 2H, Ar-*H*), 7.21 (m, 3H, Ar-*H*), 5.30 (t, *J* = 3.3 Hz, 1H, H-12), 4.32 (m, 2H, -OC*H*_2_C-), 3.63 (m, 2H, -SC*H*_2_), 3.20 (m, 3H, H-3, -NC*H*_2_), 2.88 (m, 2H, H-18, -NCH_2_CH_2_C***H***), 1.99 (m, 1H, -O*H*), 1.96 (m, 2H, -C*H*_2_), 1.88 (m, 3H, H-22, -C*H*, -C*H*_2_), 1.64 (m, 8H, 2 × -NC*H*_2_, 2 × -C*H*_2_), 1.51 (m, 3H, H-22, -C*H*, -C*H*_2_), 1.44 (m, 4H, 2 × -NC*H*_2_), 1.28 (m, 6H, 3 × -C*H*_2_), 1.17 (t, *J* = 3.6 Hz, 1H, H-9), 1.13 (s, 3H, -C*H*_3_), 1.05 (s, 1H, H-5), 0.97 (s, 3H, -C*H*_3_), 0.93 (s, 3H, -C*H*_3_), 0.90 (s, 3H, -C*H*_3_), 0.89 (s, 3H, -C*H*_3_), 0.76 (s, 3H, -C*H*_3_), 0.75 (s, 3H, -C*H*_3_). ^13^C NMR (100 MHz, CDCl_3_): δ 195.1 (-*C*S_2_), 177.5 (C-28), 144.3 (*Ph*), 143.6 (C-13), 128.7 (*Ph*), 126.8 (*Ph*), 122.6 (C-12), 78.9 (C-3), 62.4 (-*C*O-), 55.2 (C-5), 47.6 (C-9), 46.8 (C-17), 45.8 (C-19), 42.6 (-N*C*H_2_), 41.7 (C-14), 41.3 (C-18), 39.4 (C-8), 38.8 (C-1), 38.5 (C-4), 37.0 (C-10), 35.9 (-*C*S), 34.7 (-NCH_2_CH_2_***C***H), 33.9 (C-29), 33.2 (C-22), 32.8 (C-21), 32.4 (C-7), 30.7 (C-20), 28.2 (C-15), 27.7 (C-23), 27.2 (C-27), 27.0 (-NCH_2_***C***H_2_), 25.9 (C-30), 25.3 (-NCH_2_***C***H_2_), 23.7 (C-2), 23.5 (C-11), 23.0 (C-16), 18.4 (C-6), 17.2 (C-26), 15.7 (C-24), 15.4 (C-25). HRMS (ESI): *m*/*z* calculated for C_44_H_65_NO_3_S_2_ [M+H]^+^: 720.4484. Found: 720.4450.

[2-((4-methylpiperazine-1-carbonothioyl)thio)ethyl] 3-hydroxy-12-en-28-oic acid (**3n**). To a mixture of CS_2_ (1.8 mmol, 108 µL), anhydrous K_3_PO_4_ (0.8 mmol, 169.1 mg), and THF (8.0 mL), 4-methylpiperazine (1.0 mmol, 112 µL) was slowly added at 0 °C, and the reaction mixture was then stirred at 0 °C for 0.5 h. Another THF solution (4.0 mL) of **2** (0.4 mmol, 225.0 mg) was added dropwise to the resulting mixture. The reaction mixture was stirred for 12 h at room temperature, then quenched with ice water (15.0 mL), and the insoluble material was removed by a Buchner funnel. After removal of the solvent, the residue was dissolved in ethyl acetate (15.0 mL). Water (15.0 mL) was added to the resulting solution, the organic layer was separated, and the aqueous layer was extracted with ethyl acetate (15.0 mL × 2). The organic solutions were combined and dried over anhydrous Na_2_SO_4_. After removal of the solvent, the residue was submitted to column chromatography on silica gel (200–300 mesh) using petroleum ether and ethyl acetate (1/4 in *v*/*v*) as eluents to give **3n** (234.3 mg, 89% yield) as a yellowish gel. ^1^H NMR (400 MHz, CDCl_3_): 5.30 (t, *J* = 3.3 Hz, 1H, H-12), 4.36 (s, 2H, -OC*H*_2_C-), 4.28 (m, 2H, -NC*H*_2_), 3.96 (s, 2H, -NC*H*_2_), 3.61 (m, 2H, -SC*H*_2_), 3.21 (m, 1H, H-3), 2.87 (dd, *J* = 13.6, 4.0 Hz, 1H, H-18), 2.50 (s, 4H, 2 × -NC*H*_2_), 2.34 (s, 3H, -C*H*_3_), 1.97 (m, 1H, -O*H*), 1.87 (m, 2H, -C*H*_2_), 1.64 (m, 6H, 3 × -C*H*_2_), 1.53 (m, 3H, H-22, -C*H*, -C*H*_2_), 1.40 (m, 3H, H-22, -C*H*, -C*H*_2_), 1.27 (m, 6H, 3 × -C*H*_2_), 1.16 (t, *J* = 4.3 Hz, 1H, H-9), 1.13 (s, 3H, -C*H*_3_), 1.05 (s, 1H, H-5), 0.98 (s, 3H, -C*H*_3_), 0.93 (s, 3H, -C*H*_3_), 0.90 (s, 6H, 2 × -C*H*_3_), 0.78 (s, 3H, -C*H*_3_), 0.73 (s, 3H, -C*H*_3_). ^13^C NMR (100 MHz, CDCl_3_): δ 195.9 (-*C*S_2_), 177.4 (C-28), 143.6 (C-13), 122.5 (C-12), 78.9 (C-3), 62.3 (-*C*O-), 55.2 (C-5), 54.4 (-N*C*H_2_), 47.6 (C-9), 46.7 (C-17), 46.4 (-CH_2_***C***H_2_), 45.8 (C-19), 45.6 (-N*C*H_3_), 41.7 (C-14), 41.3 (C-18), 39.3 (C-8), 38.7 (C-1), 38.4 (C-4), 37.0 (C-10), 35.6 (-*C*S), 33.8 (C-29), 33.1 (C-22), 32.7 (C-21), 32.4 (C-7), 30.7 (C-20), 28.1 (C-15), 27.7 (C-23), 27.2 (C-27), 25.9 (C-30), 23.6 (C-2), 23.4 (C-11), 22.9 (C-16), 18.3 (C-6), 17.1 (C-26), 15.6 (C-24), 15.3 (C-25). HRMS (ESI): *m*/*z* calculated for C_38_H_62_N_2_O_3_S_2_ [M+H]^+^: 659.4280. Found: 659.4239.

[2-((4-(2-hydroxyethyl)piperazine-1-carbonothioyl)thio)ethyl] 3-hydroxy-12-en-28-oic acid (**3o**). To a mixture of CS_2_ (1.8 mmol, 108 µL), anhydrous K_3_PO_4_ (0.8 mmol, 169.1 mg), and THF (8.0 mL), 4-(2-hydroxyethyl)piperazine (1.0 mmol, 123 µL) was slowly added at 0 °C, and the reaction mixture was then stirred at 0 °C for 0.5 h. Another THF solution (4.0 mL) of **2** (0.4 mmol, 224.9 mg) was added dropwise to the resulting mixture. The reaction mixture was stirred for 12 h at room temperature, then quenched with ice water (15.0 mL), and the insoluble material was removed by a Buchner funnel. After removal of the solvent, the residue was dissolved in ethyl acetate (15.0 mL). Water (15.0 mL) was added to the resulting solution, the organic layer was separated, and the aqueous layer was extracted with ethyl acetate (15.0 mL × 2). The organic solutions were combined and dried over anhydrous Na_2_SO_4_. After removal of the solvent, the residue was submitted to column chromatography on silica gel (200–300 mesh) using petroleum ether and ethyl acetate (1/5 in *v*/*v*) as eluents to give **3o** (195.5 mg, 71% yield) as a yellowish gel. ^1^H NMR (400 MHz, CDCl_3_): δ 5.30 (t, *J* = 3.1 Hz, 1H, H-12), 4.33 (s, 1H, -NC*H*), 4.28 (m, 2H, -OC*H*_2_C-), 3.99 (s, 1H, -NC*H*), 3.69 (m, 2H, -NC*H*_2_), 3.61 (m, 2H, -SC*H*_2_), 3.21 (m, 1H, H-3), 2.86 (dd, *J* = 13.7, 4.1 Hz, 1H, H-18), 2.63 (t, *J* = 5.2 Hz, 4H, 2 × -NC*H*_2_), 2.61 (s, 1H, -O*H*), 1.97 (m, 1H, -O*H*), 1.88 (m, 2H, -C*H*_2_), 1.66 (m, 2H, -C*H*_2_), 1.60 (m, 5H, H-22, -C*H*, 2 × -C*H*_2_), 1.53 (m, 4H, 2 × -C*H*_2_), 1.43 (m, 3H, H-22, -C*H*, -C*H*_2_), 1.33 (m, 4H, 2 × -C*H*_2_), 1.27 (m, 4H, 2 × -C*H*_2_), 1.16 (t, *J* = 4.4 Hz, 1H, H-9), 1.13 (s, 3H, -C*H*_3_), 1.05 (s, 1H, H-5), 0.98 (s, 3H, -C*H*_3_), 0.93 (s, 3H, -C*H*_3_), 0.90 (s, 6H, 2 × -C*H*_3_), 0.77 (s, 3H, -C*H*_3_), 0.73 (s, 3H, -C*H*_3_). ^13^C NMR (100 MHz, CDCl_3_): δ 196.1 (-*C*S_2_), 177.5 (C-28), 143.6 (C-13), 122.5 (C-12), 79.0 (C-3), 62.3 (-*C*O-), 59.1 (-N***C***H_2_CH_2_OH), 57.9 (-NCH_2_***C***H_2_OH), 55.2 (C-5), 52.3 (-N***C***H_2_CH_2_), 47.6 (C-9), 46.8 (C-17), 45.8 (C-19), 41.7 (C-14), 41.3 (C-18), 39.4 (C-8), 38.8 (C-1), 38.5 (C-4), 37.0 (C-10), 35.7 (-*C*S), 33.9 (C-29), 33.1 (C-22), 32.4 (C-21), 31.5 (C-7), 30.7(-NCH_2_***C***H_2_), 30.2 (C-20), 28.1 (C-15), 27.7 (C-23), 27.2 (C-27), 25.9 (C-30), 23.7 (C-2), 23.4 (C-11), 22.9 (C-16), 18.4 (C-6), 17.1 (C-26), 15.6 (C-24), 15.4 (C-25). HRMS (ESI): *m*/*z* calculated for C_39_H_64_N_2_O_4_S_2_ [M+H]^+^: 689.4386. Found: 689.4336.

[2-((4-phenylpiperazine-1-carbonothioyl)thio)ethyl] 3-hydroxy-12-en-28-oic acid (**3p**). To a mixture of CS_2_ (1.8 mmol, 108 µL), anhydrous K_3_PO_4_ (0.8 mmol, 169.1 mg), and THF (8.0 mL), 4-phenylpiperazine (1.0 mmol, 150 µL) was slowly added at 0 °C, and the reaction mixture was then stirred at 0 °C for 0.5 h. Another THF solution (4.0 mL) of **2** (0.4 mmol, 226.7 mg) was added dropwise to the resulting mixture. The reaction mixture was stirred for 12 h at room temperature, then quenched with ice water (15.0 mL), and the insoluble material was removed by a Buchner funnel. After removal of the solvent, the residue was dissolved in ethyl acetate (15.0 mL). Water (15.0 mL) was added to the resulting solution, the organic layer was separated, and the aqueous layer was extracted with ethyl acetate (15.0 mL × 2). The organic solutions were combined and dried over anhydrous Na_2_SO_4_. After removal of the solvent, the residue was submitted to column chromatography on silica gel (200–300 mesh) using petroleum ether and ethyl acetate (1/5 in *v*/*v*) as eluents to give **3p** (247.8 mg, 86% yield) as a yellowish solid. ^1^H NMR (400 MHz, CDCl_3_): δ 7.30 (m, 2H, Ar-*H*), 6.93 (m, 3H, Ar-*H*), 5.30 (t, *J* = 3.2 Hz, 1H, H-12), 4.49 (s, 2H, -OC*H*_2_C-), 4.29 (m, 2H, -NC*H*_2_), 4.10 (m, 2H, -NC*H*_2_), 3.64 (m, 2H, -SC*H*_2_), 3.30 (t, *J* = 4.9 Hz, 4H, 2 × -NC*H*_2_), 3.20 (m, 1H, H-3), 2.87 (dd, *J* = 13.6, 3.9 Hz, 1H, H-18), 1.97 (m, 1H, -O*H*), 1.88 (m, 2H, -C*H*_2_), 1.67 (m, 4H, 2 × -C*H*_2_), 1.58 (m, 4H, 2 × -C*H*_2_), 1.51 (m, 3H, H-22, -C*H*, -C*H*_2_), 1.35 (m, 7H, H-22, -C*H*, 3 × -C*H*_2_), 1.17 (t, *J* = 4.0 Hz, 1H, H-9), 1.13 (s, 3H, -C*H*_3_), 1.05 (s, 1H, H-5), 0.97 (s, 3H, -C*H*_3_), 0.93 (s, 3H, -C*H*_3_), 0.90 (s, 3H, -C*H*_3_), 0.89 (s, 3H, -C*H*_3_), 0.76 (s, 3H, -C*H*_3_), 0.74 (s, 3H, -C*H*_3_). ^13^C NMR (100 MHz, CDCl_3_): δ 196.1 (-*C*S_2_), 177.4 (*Ph*), 150.2 (*Ph*), 143.6 (C-13), 129.3 (*Ph*), 122.5 (C-12), 120.6 (*Ph*), 116.3 (*Ph*), 78.9 (C-3), 62.2 (-*C*O-), 55.2 (C-5), 48.7 (-N*C*H_2_), 48.2 (-N*C*H_2_), 47.6 (C-9), 46.7 (C-17), 45.8 (C-19), 41.6 (C-14), 41.3 (C-18), 39.3 (C-8), 38.7 (C-1), 38.4 (C-4), 37.0 (C-10), 35.6 (-*C*S), 33.8 (C-29), 33.1 (C-22), 32.7 (C-21), 32.4 (C-7), 30.7 (C-20), 28.1 (C-15), 27.7 (C-23), 27.1 (C-27), 25.9 (C-30), 23.6 (C-2), 23.4 (C-11), 22.9 (C-16), 18.3 (C-6), 17.1 (C-26), 15.6 (C-24), 15.3 (C-25). HRMS (ESI): *m*/*z* calculated for C_43_H_64_N_2_O_3_S_2_ [M+H]^+^: 721.4437. Found: 721.4411.

[2-((4-(o-tolyl)piperazine-1-carbonothioyl)thio)ethyl] 3-hydroxy-12-en-28-oic acid (**3q**). To a mixture of CS_2_ (1.8 mmol, 108 µL), anhydrous K_3_PO_4_ (0.8 mmol, 169.1 mg), and THF (8.0 mL), 4-(o-tolyl)piperazine (1.0 mmol, 177.9 mg) was slowly added at 0 °C, and the reaction mixture was then stirred at 0 °C for 0.5 h. Another THF solution (4.0 mL) of **2** (0.4 mmol, 225.9 mg) was added dropwise to the resulting mixture. The reaction mixture was stirred for 12 h at room temperature, then quenched with ice water (15.0 mL), and the insoluble material was removed by a Buchner funnel. After removal of the solvent, the residue was dissolved in ethyl acetate (15.0 mL). Water (15.0 mL) was added to the resulting solution, the organic layer was separated, and the aqueous layer was extracted with ethyl acetate (15.0 mL × 2). The organic solutions were combined and dried over anhydrous Na_2_SO_4_. After removal of the solvent, the residue was submitted to column chromatography on silica gel (200–300 mesh) using petroleum ether and ethyl acetate (10/1 in *v*/*v*) as eluents to give **3q** (246.7 mg, 84% yield) as a yellowish gel. ^1^H NMR (400 MHz, CDCl_3_): δ 7.17 (t, *J* = 7.9 Hz, 1H, Ar-*H*), 6.73 (m, 3H, Ar-*H*), 5.30 (t, *J* = 3.1 Hz, 1H, H-12), 4.46 (s, 2H, -OC*H*_2_C-), 4.27 (m, 2H, -NC*H*_2_), 4.10 (s, 2H, -NC*H*_2_), 3.64 (m, 2H, -SC*H*_2_), 3.28 (t, *J* = 5.0 Hz, 4H, 2 × -NC*H*_2_), 3.18 (m, 1H, H-3), 2.87 (dd, *J* = 13.6, 3.8 Hz, 1H, H-18), 2.32 (s, 3H, -C*H*_3_), 1.97 (m, 1H, -O*H*), 1.87 (m, 2H, -C*H*_2_), 1.69 (m, 4H, 2 × -C*H*_2_), 1.58 (m, 5H, H-22, -C*H*, 2 × -C*H*_2_), 1.51 (m, 3H, H-22, -C*H*, -C*H*_2_), 1.35 (m, 6H, 3 × -C*H*_2_), 1.17 (t, *J* = 3.8 Hz, 1H, H-9), 1.13 (s, 3H, -C*H*_3_), 1.05 (s, 1H, H-5), 0.97 (s, 3H, -C*H*_3_), 0.93 (s, 3H, -C*H*_3_), 0.90 (s, 3H, -C*H*_3_), 0.89 (s, 3H, -C*H*_3_), 0.76 (s, 3H, -C*H*_3_), 0.74 (s, 3H, -C*H*_3_). ^13^C NMR (100 MHz, CDCl_3_): δ 195.9 (-*C*S_2_), 177.4 (C-28), 150.2 (*Ph*), 143.5 (C-13), 139.0 (*Ph*), 129.1 (*Ph*), 122.5 (C-12), 121.5 (*Ph*), 117.1 (*Ph*), 113.4 (*Ph*), 78.8 (C-3), 62.3 (-*C*O-), 55.2 (C-5), 48.8 (-N*C*H_2_), 47.6 (C-9), 46.7 (C-17), 45.8 (C-19), 41.6 (C-14), 41.3 (C-18), 39.3 (C-8), 38.7 (C-1), 38.4 (C-4), 37.0 (C-10), 35.6 (-*C*S), 33.8 (C-29), 33.1 (C-22), 32.7 (C-21), 32.4 (C-7), 30.7 (C-20), 28.1 (C-15), 27.7 (C-23), 27.1 (C-27), 26.9 (-NCH_2_***C***H_2_), 25.9 (C-30), 23.6 (C-2), 23.4 (C-11), 22.9 (C-16), 21.8 (-*C*H_3_), 18.3 (C-6), 17.1 (C-26), 15.6 (C-24), 15.4 (C-25). HRMS (ESI): *m*/*z* calculated for C_44_H_66_N_2_O_3_S_2_ [M+H]^+^: 735.4593. Found: 735.4540.

[2-((4-(m-tolyl)piperazine-1-carbonothioyl)thio)ethyl] 3-hydroxy-12-en-28-oic acid (**3r**). To a mixture of CS_2_ (1.8 mmol, 108 µL), anhydrous K_3_PO_4_ (0.8 mmol, 169.1 mg), and THF (8.0 mL), 4-(m-tolyl)piperazine (1.0 mmol, 174 µL) was slowly added at 0 °C, and the reaction mixture was then stirred at 0 °C for 0.5 h. Another THF solution (4.0 mL) of **2** (0.4 mmol, 226.1 mg) was added dropwise to the resulting mixture. The reaction mixture was stirred for 12 h at room temperature, then quenched with ice water (15.0 mL), and the insoluble material was removed by a Buchner funnel. After removal of the solvent, the residue was dissolved in ethyl acetate (15.0 mL). Water (15.0 mL) was added to the resulting solution, the organic layer was separated, and the aqueous layer was extracted with ethyl acetate (15.0 mL × 2). The organic solutions were combined and dried over anhydrous Na_2_SO_4_. After removal of the solvent, the residue was submitted to column chromatography on silica gel (200–300 mesh) using petroleum ether and ethyl acetate (10/1 in *v*/*v*) as eluents to give **3r** (235.0 mg, 80% yield) as a white solid. ^1^H NMR (400 MHz, CDCl_3_): δ 7.16 (t, *J* = 7.9 Hz, 1H, Ar-*H*), 6.72 (m, 3H, Ar-*H*), 5.30 (s, 1H, H-12), 4.40 (s, 2H, -OC*H*_2_C-), 4.28 (m, 2H, -NC*H*_2_), 4.11 (m, 2H, -NC*H*_2_), 3.60 (m, 2H, -SC*H*_2_), 3.27 (m, 4H, 2 × -NC*H*_2_), 3.18 (m, 1H, H-3), 2.87 (dd, *J* = 13.5, 3.5 Hz, 1H, H-18), 2.32 (s, 3H, -C*H*_3_), 1.95 (m, 1H, -O*H*), 1.86 (m, 2H, -C*H*_2_), 1.62 (m, 7H, H-22, -C*H*, 3 × -C*H*_2_), 1.51 (m, 3H, H-22, -C*H*, -C*H*_2_), 1.42 (m, 2H, -C*H*_2_), 1.32 (m, 6H, 3 × -C*H*_2_), 1.17 (t, *J* = 3.1 Hz, 1H, H-9), 1.13 (s, 3H, -C*H*_3_), 1.04 (s, 1H, H-5), 0.96 (s, 3H, -C*H*_3_), 0.93 (s, 3H, -C*H*_3_), 0.90 (s, 3H, -C*H*_3_), 0.89 (s, 3H, -C*H*_3_), 0.75 (s, 3H, -C*H*_3_), 0.74 (s, 3H, -C*H*_3_). ^13^C NMR (100 MHz, CDCl_3_): δ 195.9 (-*C*S_2_), 177.4 (C-28), 150.2 (*Ph*), 143.5 (C-13), 139.0 (*Ph*), 129.1 (*Ph*), 122.5 (C-12), 121.5 (*Ph*), 117.1 (*Ph*), 113.4 (*Ph*), 78.8 (C-3), 62.3 (-*C*O-), 55.2 (C-5), 48.8 (-N*C*H_2_), 47.6 (C-9), 46.7 (C-17), 45.8 (C-19), 41.6 (C-14), 41.3 (C-18), 39.3 (C-8), 38.7 (C-1), 38.4 (C-4), 37.0 (C-10), 35.6 (-*C*S), 33.8 (C-29), 33.1 (C-22), 32.7 (C-21), 32.4 (C-7), 30.7 (C-20), 28.1 (C-15), 27.7 (C-23), 27.1 (C-27), 26.9 (-NCH_2_*C*H_2_), 25.9 (C-30), 23.6 (C-2), 23.4 (C-11), 22.9 (C-16), 21.8 (-CH_3_), 18.3 (C-6), 17.1 (C-26), 15.6 (C-24), 15.4 (C-25). HRMS (ESI): *m*/*z* calculated for C_44_H_66_N_2_O_3_S_2_ [M+H]^+^: 735.4593. Found: 735.4537.

[2-((4-(p-tolyl)piperazine-1-carbonothioyl)thio)ethyl] 3-hydroxy-12-en-28-oic acid (**3s**). To a mixture of CS_2_ (1.8 mmol, 108 µL), anhydrous K_3_PO_4_ (0.8 mmol, 169.1 mg), and THF (8.0 mL), 4-(p-tolyl)piperazine (1.0 mmol, 177.1 mg) was slowly added at 0 °C, and the reaction mixture was then stirred at 0 °C for 0.5 h. Another THF solution (4.0 mL) of **2** (0.4 mmol, 225.8 mg) was added dropwise to the resulting mixture. The reaction mixture was stirred for 12 h at room temperature, then quenched with ice water (15.0 mL), and the insoluble material was removed by a Buchner funnel. After removal of the solvent, the residue was dissolved in ethyl acetate (15.0 mL). Water (15.0 mL) was added to the resulting solution, the organic layer was separated, and the aqueous layer was extracted with ethyl acetate (15.0 mL × 2). The organic solutions were combined and dried over anhydrous Na_2_SO_4_. After removal of the solvent, the residue was submitted to column chromatography on silica gel (200–300 mesh) using petroleum ether and ethyl acetate (10/1 in *v*/*v*) as eluents to give **3s** (249.6 mg, 85% yield) as a white solid. ^1^H NMR (400 MHz, CDCl_3_): δ 7.10 (d, *J* = 8.3 Hz, 2H, Ar-*H*), 6.84 (d, *J* = 8.5 Hz, 2H, Ar-*H*), 5.30 (t, *J* = 3.3 Hz, 1H, H-12), 4.48 (s, 2H, -NC*H*_2_), 4.30 (m, 2H, -OC*H*_2_C-), 4.10 (s, 2H, -NC*H*_2_), 3.64 (m, 2H, -SC*H*_2_), 3.23 (m, 4H, 2× -NC*H*_2_), 3.19 (d, *J* = 4.8 Hz, 1H, H-3), 2.87 (dd, *J* = 13.7, 4.0 Hz, 1H, H-18), 2.28 (s, 3H, -C*H*_3_), 1.97 (m, 1H, -O*H*), 1.87 (m, 2H, 2× -C*H*_2_), 1.65 (m, 6H, 3× -C*H*_2_), 1.50 (m, 4H, 2× -C*H*_2_), 1.40 (m, 4H, 2× -C*H*_2_), 1.29 (m, 4H, 2× -C*H*_2_) 1.17 (t, *J* = 4.1 Hz, 1H, H-9), 1.13 (s, 3H, -C*H*_3_), 1.05 (s, 1H, H-5), 0.97 (s, 3H, -C*H*_3_), 0.93 (s, 3H, -C*H*_3_), 0.90 (s, 3H, -C*H*_3_), 0.89 (s, 3H, -C*H*_3_), 0.76 (s, 3H, -C*H*_3_), 0.74 (s, 3H, -C*H*_3_). ^13^C NMR (100 MHz, CDCl_3_): δ 196.0 (-*C*S_2_), 177.4 (C-28), 148.1 (*Ph*), 143.6 (C-13), 130.3 (*Ph*), 129.8 (*Ph*), 122.5 (C-12), 116.8 (*Ph*), 78.9 (C-3), 62.3 (-*C*O-), 55.2 (C-5), 49.4 (-N*C*H_2_), 47.6 (C-9), 46.7 (C-17), 45.8 (C-19), 41.7 (C-14), 41.5 (C-18), 41.3 (-NCH_2_***C***H_2_), 39.3 (C-8), 38.7 (C-1), 38.4 (C-4), 37.0 (C-10), 35.6 (-CS), 33.8 (C-29), 33.1 (C-22), 32.7 (C-21), 32.4 (C-7), 30.7 (C-20), 28.1 (C-15), 27.7 (C-23), 27.1 (C-27), 25.9 (C-30), 23.6 (C-2), 23.4 (C-11), 22.9 (C-16), 20.5 (-*C*H_3_), 18.3 (C-6), 17.1 (C-26), 15.6 (C-24), 15.3 (C-25). HRMS (ESI): *m*/*z* calculated for C_44_H_66_N_2_O_3_S_2_ [M+H]^+^: 735.4593. Found: 735.4562.

[2-((thiomorpholine-4-carbonothioyl)thio)ethyl] 3-hydroxy-12-en-28-oic acid (**3t**). To a mixture of CS_2_ (1.8 mmol, 108 µL), anhydrous K_3_PO_4_ (0.8 mmol, 169.1 mg), and THF (8.0 mL), thiomorpholine (1.0 mmol, 94 µL) was slowly added at 0 °C, and the reaction mixture was then stirred at 0 °C for 0.5 h. Another THF solution (4.0 mL) of **2** (0.4 mmol, 225.1 mg) was added dropwise to the resulting mixture. The reaction mixture was stirred for 12 h at room temperature, then quenched with ice water (15.0 mL), and the insoluble material was removed by a Buchner funnel. After removal of the solvent, the residue was dissolved in ethyl acetate (15.0 mL). Water (15.0 mL) was added to the resulting solution, the organic layer was separated, and the aqueous layer was extracted with ethyl acetate (15.0 mL × 2). The organic solutions were combined and dried over anhydrous Na_2_SO_4_. After removal of the solvent, the residue was submitted to column chromatography on silica gel (200–300 mesh) using petroleum ether and ethyl acetate (10/1 in *v*/*v*) as eluents to give **3t** (190.4 mg, 72% yield) as a white gel. ^1^H NMR (400 MHz, CDCl_3_): δ 5.29 (s, 1H, H-12), 4.61 (s, 2H, -OC*H*_2_C-), 4.26 (m, 2H, -NC*H*_2_), 3.61 (m, 2H, -SC*H*_2_), 3.21 (m, 1H, H-3), 2.86 (dd, *J* = 13.5, 3.6 Hz, 1H, H-18), 2.75 (m, 4H, 2 × -SC*H*_2_), 1.97 (m, 1H, -O*H*), 1.88 (m, 2H, -C*H*_2_), 1.63 (m, 6H, 3 × -C*H*_2_), 1.53 (m, 3H, H-22, -C*H*, -C*H*_2_), 1.42 (m, 4H, 2 × -C*H*_2_), 1.35 (m, 3H, H-22, -C*H*, -C*H*_2_), 1.27 (m, 4H, 2 × -C*H*_2_), 1.16 (t, *J* = 3.6 Hz, 1H, H-9), 1.13 (s, 3H, -C*H*_3_), 1.04 (s, 1H, H-5), 0.98 (s, 3H, -C*H*_3_), 0.93 (s, 3H, -C*H*_3_), 0.90 (s, 6H, 2 × -C*H*_3_), 0.78 (s, 3H, -C*H*_3_), 0.73 (s, 3H, -C*H*_3_). ^13^C NMR (100 MHz, CDCl_3_): δ 195.9 (-*C*S_2_), 177.5 (C-28), 143.6 (C-13), 122.6 (C-12), 79.0 (C-3), 62.2 (-*C*O-), 55.3 (C-5), 47.7 (C-9), 46.8 (C-17), 45.8 (C-19), 41.7 (C-14), 41.3 (C-18), 39.4 (C-8), 38.8 (C-1), 38.5 (C-4), 37.1 (C-10), 35.8 (-*C*S), 33.9 (C-29), 33.2 (C-22), 32.8 (C-21), 32.5 (C-7), 30.8 (C-20), 28.2 (C-15), 27.8 (C-23), 27.2 (C-27), 27.0 (-N*C*H_2_), 25.9 (C-30), 23.7 (C-2), 23.5 (C-11), 23.0 (C-16), 18.4 (C-6), 17.2 (C-26), 15.7 (C-24), 15.4 (C-25), 14.2 (-SCH_2_). HRMS (ESI): *m*/*z* calculated for C_37_H_59_NO_3_S_3_ [M+H]^+^: 662.3735. Found: 662.3673.

### 3.3. Preliminary Biological Study

The in vitro cytotoxic activities of the compounds were evaluated by MTT assay against Panc1, A549, Hep3B, Huh-7, HT-29, Hela, LO2. Cell lines were obtained from the Laboratory of Molecular Pharmacology, Southwest Medical University. Briefly, different tumor cells grew in DMEM medium except for A549, which used 1640 medium. Cells ((3–5) × 10^3^ cells/well) were harvested at the log phase of growth and seeded in 96-well plates. After 24 h incubation at 37 °C in 5% CO_2_ to allow cell attachment, cultures were exposed to various concentrations of the isolated compounds for 48 h. Finally, the MTT solution was added. Plates were further incubated for 4 h at 37 °C after adding 150 µL/well of DMSO and shaking for 10 min on the shaker platform. The plates were read in a 96-well plate reader at 490 nm wavelength. The results were expressed as IC_50_ values, and were defined as the concentration at which 50% survival of cells was obtained. Fluorouracil, docetaxel, and cisplatin were co-assayed as positive controls.

## 4. Conclusions

In summary, we have synthesized a series of OA-dithiocarbamate derivatives in a two-step protocol at room temperature, offering a readily accessible synthetic route to obtain novel OA derivatives in high yields. Moreover, some of the compounds were shown to be promising hit compounds, with remarkably improved broad-spectrum antiproliferative activities compared to the natural product OA. Mechanistic insights of their activities on certain tumor cell lines are currently underway in our laboratory.

## Data Availability

Not applicable.

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
