# Peer review of "Synthesis of Oleanolic Acid-Dithiocarbamate Conjugates and Evaluation of Their Broad-Spectrum Antitumor Activities"

_molecules, 2023, doi:10.3390/molecules28031414_

Round 1

Reviewer 1 Report

The manuscript "Synthesis of Oleanolic Acid-Dithiocarbamate Conjugates and 2 Evaluation of Their Broad-Spectrum Antitumor Activities" by L. Tang et al. describes a two-step procedure for the synthesis of a novel series of oleanolic acid-dithiocarbamate conjugates and their antiproliferative evaluation against a number of cancer cell lines and one normal cell line. The reaction conditions for the synthesis of these conjugates were optimized and very well described in the text. English is refreshingly good, only minor spell check is needed. However, some changes to the manuscript are necessary before publication.

1. The sentence in the Introduction: "Extensive research has revealed that pentacyclic triterpenes are the main active ingredients of many traditional Chinese medicines." is identical to the sentence in ref. 33. and should be rewritten.

2. In Schemes 1 and 2, authors should add references next to each compound.

3. In the Results and Discussion section, authors should discuss NMR spectra and how they identified the structures of the newly synthesized compounds.

4. MS spectra should be added in the Supplementary file.

5. In the Materials and Methods, it would be better to describe the general procedure for the synthesis of 3a-t than to describe it only for the synthesis of 3a. Otherwise, the sentence "The synthesis of 3b was conducted like that of 3a (213.6mg, 83% yield) as a white solid." should be rewritten as "... to give 3b... as a white solid". This goes for all the following compounds. Additionally, for all the following compounds, starting with 3b, authors should give the volume or mass of amine used in the synthesis.

6. Line 134: "was using" should be replaced by "...was measured using...".

7. In Table 3., all active compounds should be highlighted. It is not clear why only 3e is chosen as no other experiments besides antiproliferative evaluation were conducted.

8. What is the concentration range used for IC50 determination, and how many concentrations were tested?

9. How many replicates of each concentration were tested and in how many experiments?

10. Additionally, the authors should at least calculate the molecular properties (i.e. clogP, TPSA, HBA, HBD etc.). Otherwise, knowing only preliminary antiproliferative activities, the compounds are far from being leads (just unvalidated hits). Furthermore, the MW of the compounds is rather large >600 and they could hardly serve as good lead compounds, so authors should rewrite this statement.

11. If possible, the authors should conduct some additional biological experiments (e.g. cell morphology study, cell cycle analysis, possible induction of apoptosis).

12. The term "anti-hepatodamage" should be replaced by "anti-hepatotoxic".

Author Response

We would like to take this chance to thank the referees very much for efficient editing and reviewing this manuscript. Your valuable comments have certainly made a much better presentation of this paper. I trust these revisions fairly address the points raised by the reviewers and sincerely hope that this revised manuscript is now acceptable for publication. Thank you.

  1. The sentence in the Introduction: "Extensive research has revealed that pentacyclic triterpenes are the main active ingredients of many traditional Chinese medicines." is identical to the sentence in ref. 33. and should be rewritten.

ANS: As per suggestion, the sentence has been rewritten as “Recently, pentacyclic triterpenes have been identified as the main biological active components in many traditional Chinese medicines.”

  1. In Schemes 1 and 2, authors should add references next to each compound.

ANS: As per suggestion, references have been added for each compound in Schemes 1 and 2.

  1. In the Results and Discussion section, authors should discuss NMR spectra and how they identified the structures of the newly synthesized compounds.

ANS: As per suggestion, the spectra data of each compound has been completely interpreted.

  1.  MS spectra should be added in the Supplementary file.

ANS: As per suggestion, the MS spectra has been added in the Supplementary file.

  1.  In the Materials and Methods, it would be better to describe the general procedure for the synthesis of 3a-tthan to describe it only for the synthesis of 3a. Otherwise, the sentence "The synthesis of 3bwas conducted like that of 3a (213.6mg, 83% yield) as a white solid." should be rewritten as "... to give 3b... as a white solid". This goes for all the following compounds. Additionally, for all the following compounds, starting with 3b, authors should give the volume or mass of amine used in the synthesis.

ANS: As per suggestion, the statement has been rewritten as “the general procedure for the synthesis of 3a-t” in the Materials and Methods.  Additionally, the volume or mass of amines used in the synthesis of all compounds has been added in the manuscript.

  1. Line 134: "was using" should be replaced by "...was measured using...".

ANS: As per suggestion, the statement has been rewritten as “...was measured using...” in the Line 134.

  1. In Table 3., all active compounds should be highlighted. It is not clear why only 3e is chosen as no other experiments besides antiproliferative evaluation were conducted.

ANS: As per suggestion, all active compounds (IC50 <10 uM) have been highlighted.

  1. What is the concentration range used for IC50determination, and how many concentrations were tested?

ANS: ANS: The concentration range used for IC50 determination is 0-80 µM, the concentration gradients were 0 µM, 10 µM, 20 µM, 40 µM, 60 µM and 80 µM.

  1. How many replicates of each concentration were tested and in how many experiments?

ANS: Each concentration was tested for three replicates and three experiments, and the final data in the table was averaged.

  1. Additionally, the authors should at least calculate the molecular properties (i.e. clogP, TPSA, HBA, HBD etc.). Otherwise, knowing only preliminary antiproliferative activities, the compounds are far from being leads (just unvalidated hits). Furthermore, the MW of the compounds is rather large >600 and they could hardly serve as good lead compounds, so authors should rewrite this statement.

ANS: As per suggestion, the statement of “lead compounds” has been rewritten as “hit compounds”. Calculation of the molecular properties is difficult to achieve within 10 days, because currently we are affected by the COVID-19 pandemic.

  1. If possible, the authors should conduct some additional biological experiments (e.g. cell morphology study, cell cycle analysis, possible induction of apoptosis).

ANS: Additional biological experiments are difficult to achieve within 10 days, because currently we are affected by the COVID-19 pandemic.

  1. The term "anti-hepatodamage" should be replaced by "anti-hepatotoxic".

ANS: As per suggestion, the term "anti-hepatodamage" has been replaced by "anti-hepatotoxic".

Reviewer 2 Report

In this contribution Liyao Tang, et. al., describe an efficient and mild synthetic route for bioactive natural product derivatives of novel oleanolic acid (OA)-dithiocarbamate conjugates. Likewise, the authors made biological evaluation of the new compounds. The motivation of the following manuscript is clear, however there are details that can be improved.

The authors describe that the more potent biological activity is due to the presence of the hydroxyl group in the OA structure. However, the authors do not say anything about why compound 3o and 3k, having hydroxyl groups in their structures, do not present potent biological responses. Perhaps, they should answer this by comparing the structures obtained.

Author Response

We would like to take this chance to thank the referees very much for efficient editing and reviewing this manuscript. Your valuable comments have certainly made a much better presentation of this paper. I trust these revisions fairly address the points raised by the reviewers and sincerely hope that this revised manuscript is now acceptable for publication. Thank you.

  1. The authors describe that the more potent biological activity is due to the presence of the hydroxyl group in the OA structure. However, the authors do not say anything about why compound 3o and 3k, having hydroxyl groups in their structures, do not present potent biological responses. Perhaps, they should answer this by comparing the structures obtained.

ANS: The relative positions of hydroxyalkyl (3k vs 3l) and extra nitrogen substitution (3o vs 3l) may affect the biological activities by different molecular mode of action. Further experiments are underway in our lab.

Reviewer 3 Report

This manuscript from Tang et al. describes the synthesis and antitumor activity evaluation of a series of oleanolic acid-dithiocarbamate Conjugates.

The natural pentacyclic triterpenoid oleanolic acid (OA) is widely distributed in food and plants. OA and its derivatives have been widely investigated for their diverse biological activities. On the other hand, dithiocarbamate is a biologically important structural motif in medicinal chemistry.  Based on the pharmacophore hybrid strategy, the authors report a two-step protocol to obtain a series of structurally novel oleanolic acid (OA)-dithiocarbamate conjugates in mild conditions and high yields.

Moreover, some of the compounds were shown remarkably improved broad-spectrum antiproliferative activities compared to the natural product OA.

I consider this work attracts sufficient readership for Molecules and thus recommend it for publication, provided that the following points are reflected in the revision process.

1.      In page 1, line 10, “without affecting normal cells” is inappropriate. Maybe it should be rewritten such as “with low cytotoxicity on normal cells”.

2.      In page 5, line 121, “In Vitro” should be in italics

3.      In page 10, 4. Conclusions  line 385-6, “in a one-pot manner” would be “in a two-step protocol”.

4.      The volume and page number of ref_31 should be provided, instead of doi.

Author Response

We would like to take this chance to thank the referees very much for efficient editing and reviewing this manuscript. Your valuable comments have certainly made a much better presentation of this paper. I trust these revisions fairly address the points raised by the reviewers and sincerely hope that this revised manuscript is now acceptable for publication. Thank you.

  1. In page 1, line 10, “without affecting normal cells” is inappropriate. Maybe it should be rewritten such as “with low cytotoxicity on normal cells”.

ANS: As per suggestion, “without affecting normal cells” has been rewritten as “with low cytotoxicity on normal cells”.

  1. In page 5, line 121, “In Vitro” should be in italics.

ANS: As per suggestion, in page 5, line 121, “In Vitro” has been changed as italics.

  1. In page 10, 4. Conclusions line 385-6, “in a one-pot manner” would be “in a two-step protocol”.

ANS: As per suggestion, in page 10, 4. Conclusions line 385-6, “in a one-pot manner” has been rewritten as “in a two-step protocol”.

  1. In page 10, 4. Conclusions line 385-6, “in a one-pot manner” would be “in a two-step protocol”.

ANS: As per suggestion, the volume and page number of ref_31 have been provided.

Round 2

Reviewer 1 Report

Thank you for accepting most of the comments and improving the article. I recommend this article for publication after minor technical corrections. In Scheme 1, "oxidation" and "addition" as well as "esterification" and "amidation" are overlapping. Additionally, please correct the text above the arrows in Tables 1 and 2.

Author Response

We would like to take this chance to thank you very much for efficient editing and reviewing this manuscript. Your valuable comments have certainly made a much better presentation of this paper. Thank you.

COMMENTS FROM THE REVIEWER:

Reviewer #1:

  1. Thank you for accepting most of the comments and improving the article. I recommend this article for publication after minor technical corrections. In Scheme 1, "oxidation" and "addition" as well as "esterification" and "amidation" are overlapping. Additionally, please correct the text above the arrows in Tables 1 and 2.

ANS: As per suggestion, (1) in Scheme 1, the structural modifications of OA has been simplified as  “Oxidation, Addition”, “Oximation, Fusing with ring-A”, and “Esterification, Amidation” on different positions of OA skeleton, respectively. (2) The text above the arrows in Tables 1 and 2 has been corrected, please see the revised manuscript (with tracked changes).